# Oncolytic virus-derived type I interferon restricts CAR T cell therapy

Laura Evgin[1], Amanda L. Huff[1], Phonphimon Wongthida[1], Jill Thompson[1], Tim Kottke[1], Jason Tonne[1], Matthew Schuelke [2], Katayoun Ayasoufi[2], Christopher B. Driscoll[1], Kevin G. Shim[2], Pierce Reynolds[1], Dileep D. Monie [1], Aaron J. Johnson[2], Matt Coffey[3], Sarah L. Young[4], Gary Archer[5], John Sampson[5], Jose Pulido[6], Luis Sanchez Perez[5] & Richard Vile [1,2 ✉]

The application of adoptive T cell therapies, including those using chimeric antigen receptor (CAR)-modified T cells, to solid tumors requires combinatorial strategies to overcome immune suppression associated with the tumor microenvironment. Here we test whether the inflammatory nature of oncolytic viruses and their ability to remodel the tumor microenvironment may help to recruit and potentiate the functionality of CAR T cells. Contrary to our hypothesis, VSVmIFNβ infection is associated with attrition of murine EGFRvIII CAR T cells in a B16EGFRvIII model, despite inducing a robust proinflammatory shift in the chemokine profile. Mechanistically, type I interferon (IFN) expressed following infection promotes apoptosis, activation, and inhibitory receptor expression, and interferon-insensitive CAR T cells enable combinatorial therapy with VSVmIFNβ. Our study uncovers an unexpected mechanism of therapeutic interference, and prompts further investigation into the interaction between CAR T cells and oncolytic viruses to optimize combination therapy.

[1] Department of Molecular Medicine, Mayo Clinic, Rochester, MN, USA. [2] Department of Immunology, Mayo Clinic, Rochester, MN, USA. [3] Oncolytics Biotech Incorporated, Calgary, Canada. [4] Department of Pathology, Dunedin School of Medicine, University of Otago, Dunedin, New Zealand. [5] Department of Neurosurgery, Duke University, Durham, NC, USA. [6] Department of Ophthalmology, Mayo Clinic, Rochester, MN, USA. ✉email: vile.richard@mayo.edu

Chimeric antigen receptor (CAR)-modified T cells have been remarkably successful in treating hematologic malignancies[1,2]. However, multiple immune suppressive mechanisms limit their infiltration and drive their dysfunction in the hostile microenvironment of solid tumors. Insufficient migration of CAR T cells into the tumor reflects an unfavorable chemokine gradient poor in CXCL9, 10, and 11, the ligands for activated T cells expressing CXCR3[2]. Suboptimal CAR T cell persistence and function in the tumor microenvironment (TME) further derives from exposure to inhibitory soluble mediators such as TGFβ, IL10, and adenosine; interaction with suppressive immune populations such as regulatory T cells, myeloid-derived suppressor cells (MDSCs), and tumor-associated macrophages; and engagement of inhibitory ligands that promote exhaustion and or apoptosis of T cells[3].

Oncolytic viruses (OVs) are naturally selected or engineered viruses which replicate preferentially in cancer cells, but which are restricted in normal healthy cells. OVs initiate targeted infection and lysis of tumor beds while simultaneously expressing therapeutic transgenes in the tumor milieu such as cytokines, tumor antigens, checkpoint inhibitors, or T cell engagers to augment anti-tumor immunity. Their nucleic acid directly activates Toll-like receptors (TLRs) and innate immune response pathways to initiate a proinflammatory cascade that stimulates the production of chemokines and cytokines including CXCL9, 10, 11, and CCL5, to alter the balance of inhibitory and activating immune cells[4–6]. In this way, multiple OVs have been shown pre-clinically and clinically at various time points after virus administration to promote T cell infiltration, PD-L1 upregulation, and render immunologically cold tumors hot[7–9]. These attributes have prompted the evaluation of oncolytic adenoviruses in combination with human CAR T cells in xenograft models[10–13].

The mechanism of tumor tropism of many OVs, and vesicular stomatitis virus (VSV) in particular, is predominantly mediated by the exquisite sensitivity of the virus to type I interferon (IFN), and the parallel loss of IFN responsiveness of tumor cells[14]. In order to increase the safety, specificity, and therapeutic index of VSV, IFNβ is encoded between the glycoprotein and polymerase genes[15,16]. Type I IFNs can additionally act as a key signal 3 cytokine to facilitate the priming of virus and tumor reactive T cells[17–19]. Therefore, the therapeutic value of VSVIFNβ lies both in its ability to induce oncolysis, as well as its ability to re-engage immune surveillance[16]. Preclinical testing of VSV encoding species-specific IFNβ[20] has led to the clinical evaluation of the platform initially in hepatocellular carcinoma, and now in diverse indications, both as a monotherapy and in combination with checkpoint inhibitors.

We hypothesize that OVs, and VSVmIFNβ in particular, have intrinsic properties that activate a proinflammatory cascade that alters the cytokine and chemokine profile of the tumor, as well as engineered properties such as the expression of therapeutic transgenes, that together will promote CAR T cells trafficking to the tumor and reduce their negative regulation. However, using a fully immunocompetent model of murine EGFRvIII third generation CAR T cells, we uncover an unexpected mechanism of antagonism whereby VSVmIFNβ infection of the tumor is associated with profound attrition of CAR T cells. Herein we show that type I IFN drives apoptosis of both conventional and CAR modified T cells via a T cell-intrinsic mechanism. We also report a CAR T cell-specific mechanism where type I IFN upregulates the CAR molecule on the surface, promotes antigen independent activation, and drives expression of inhibitory receptors. IFNAR1 KO CAR T cells outcompete IFN receptor-competent cells in vivo, and provide enhanced therapy against B16EGFRvIII tumors in a combination setting. Therefore, our data show that OV infection remodels the TME in complex ways that are both helpful and deleterious to CAR T cell therapy and further

highlights that empirical testing in fully immunocompetent models provides critical predictive information about the interaction between these two therapeutic platforms.

## Results

**VSVmIFNβ elicits chemokines which recruit activated T cells.** We first established a B16 subcutaneous model that over-expressed the CAR target antigen EGFRvIII (Supplementary Fig. 1A) and evaluated how VSVmIFNβ infection altered the cytokine and chemokine profile of the TME (Fig. 1a). Although this system utilizes a CAR T target that is atypically expressed on melanoma, our primary goal was to characterize the interactions between the virus and CAR T cells in a fully immunocompetent animal model. A single intratumoral (IT) injection induced high level expression of the transgene IFNβ that peaked between 6 and 24 h post infection (hpi) and was sustained out to 48hpi (Fig. 1b, c). IFNα was also induced along with the inflammatory chemokines CXCL10, and CCL5, and their expression mirrored the kinetics of viral replication in the tumor in vivo and in vitro (Fig. 1b, c, Supplementary Fig. 1B).

**Combination therapy does not provide superior tumor control.** Given the rapid induction of chemokines in the infected tumors that we hypothesized would recruit CXCR3 expressing CAR T cells (Supplementary Fig. 1C), we administered VSVmIFNβ 6 h prior to adoptive transfer of third generation murine EGFRvIII CAR T cells[21] to determine whether OV remodeling of the tumor could potentiate anti-tumor CAR T cell therapy (Fig. 1d). Although we observed a modest reduction in tumor growth rate with each modality, we did not observe a significant therapeutic benefit associated with the combination of VSVmIFNβ and EGFRvIII CAR T cells (Fig. 1e). In addition to the lack of improved therapy associated with the combination arm, we performed submandibular bleeds 7 days post-adoptive transfer to quantify circulating CAR T cells and observed a 3-fold reduction in the number of CD8 CAR T cells (Fig. 1f).

We repeated the same schedule of VSVmIFNβ and EGFRvIII CAR T cell mono- or combination therapy in mice that received a lymphodepleting dose of 5 Gy whole body irradiation to fully capture the therapeutic potential of the CAR T cells (Fig. 1g). Lymphodepletion was confirmed by submandibular bleeds 2 days post-irradiation (Supplementary Fig. 1D). Although the magnitude of the therapeutic effect of each modality was improved relative to non-preconditioned mice, we again did not observe any improvement in overall survival or tumor control with the combination approach (Fig. 1h). Similarly, fewer CD8 CAR T cells were detected in the blood of mice which received a single dose or three doses of VSVmIFNβ, suggesting that the VSVmIFNβ mediated attrition could not be overcome by lymphodepletion (Fig. 1i). The reduction in the number of circulating CAR T cells could not be explained by direct infection with VSV. In contrast to its oncolytic effect on the B16EGFRvIII cells, VSV did not productively infect CAR T cells as evidenced by the lack of GFP expression from VSVGFP. We also confirmed that VSVmIFNβ did not directly promote apoptosis of CAR T cells in vitro (Supplementary Fig. 1E); however, it is not possible to rule out that the level of infection of CAR T cells may be different in vitro and in vivo. EGFRvIII expression was largely maintained in B16EGFRvIII tumors which escaped CAR T therapy, suggesting that antigen loss was not a primary mechanism of therapeutic failure in this model (Supplementary Fig. 2).

**OVs promote attrition of adoptively transferred T cells.** We next sought to distinguish between VSV-mediated CAR T cell apoptosis, a block in proliferation, and migration to other parts of the body, and further to determine if the effect was T cell intrinsic

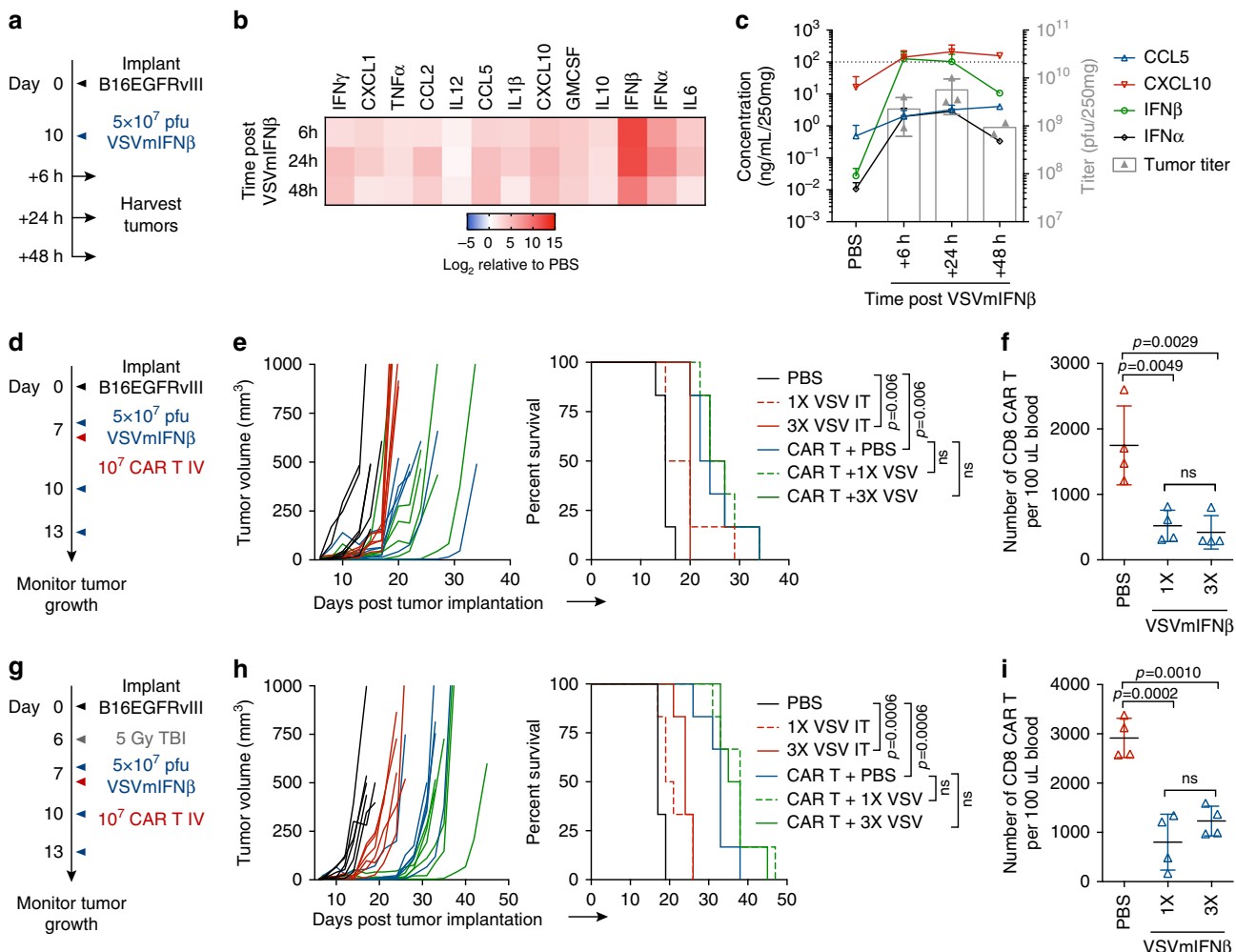

**Fig. 1 Oncolytic VSVmIFNβ infection promotes a favorable chemokine profile for CAR T cell trafficking, yet combination does not improve survival.**
Mice bearing B16EGFRvIII tumors received a single intratumoral (IT) injection of $5 \times 10^7$ pfu of VSVmIFNβ or PBS and tumors were harvested 6, 24 or 48 h post injection (**a**). Cytokine and chemokine concentration and infectious viral titer (**b**, **c**) was quantified in tumor homogenate and normalized to the average tumor weight. Mean ± SD for $n = 3$ per group except 48 h time point $n = 2$. Dashed line indicates upper limit of standard curve for quantification. **d** Mice bearing B16EGFRvIII tumors were treated with $5 \times 10^7$ pfu VSVmIFNβ or PBS 6 h prior to intravenous (IV) administration of $1 \times 10^7$ EGFRvIII CAR T cells on day 7. Select groups received 2 additional doses of $5 \times 10^7$ pfu VSVmIFNβ on days 10 and 13. $n = 6$/group. **e** Tumor growth for select groups is shown in the left panel and overall survival is shown in the right panel. **f** Circulating CD8 CAR T cells were quantified from the blood 7 days after adoptive transfer and represented as mean ± SD for $n = 4$/group. Each symbol represents a mouse. **g** Mice bearing B16EGFRvIII tumors were given a lymphodepleting dose of 5 Gy total body irradiation (TBI) on day 6 and treated with $5 \times 10^7$ pfu VSVmIFNβ or PBS 6 h prior to administration of $1 \times 10^7$ EGFRvIII CAR T cells on day 7. Select groups received 2 additional doses of $5 \times 10^7$ pfu VSVmIFNβ on days 10 and 13. $n = 6$/group. **h** Tumor growth for select groups is shown in the left panel and overall survival is shown in the right panel. Circulating CD8 CAR T cells were quantified from the blood 6 days after adoptive transfer and represented as mean ± SD for $n = 4$/group (**i**). Experiments were performed once. *P*-values were determined using the Log-rank Mantel–Cox test (**e**, **h**) or a one-way ANOVA with a Tukey multiple comparisons post-test (**f**, **i**). Statistical significance set at $p < 0.05$, ns > 0.05. Source data are provided in the Source Data File.

or CAR T cell specific. We administered a single dose of VSVmIFNβ at various time points ranging between 6 and 72 h prior to co-injection of a 1:1 mixture of CAR T cells and activated Pmel T cells, which recognize the melanoma-associated antigen gp100 (Fig. 2a, b). Three days post T cell administration, we enumerated the total viable CAR T and Pmel T cells in the tumor, spleen, lung, bone marrow, and liver (Fig. 2c–i). Fewer Pmel T cells, and CD8 and CD4 CAR T cells were recovered from tumors injected with VSVmIFNβ 6 or 24 h prior to adoptive transfer compared to those injected with PBS. A similar reduction in the endogenous CD8 compartment was observed when VSVmIFNβ was administered. We further observed a modest reduction in the number of Pmel and CD8 and CD4 CAR T cells in the spleen. However, no changes in adoptively transferred cells in the lungs,

liver or bone marrow, were observed suggesting that the reduction in CAR T and Pmel T cells in the tumor was not due to mislocalization. We hypothesized that the proliferation of the adoptively transferred T cells may be compromised due to competition with endogenous virus-responding cells for cytokines and metabolites which would account for their reduction. However, a defect in proliferation was not detected (Fig. 2j). In fact, proliferation of both the CAR T cells and Pmel T cells was enhanced when VSVmIFNβ was administered 6 h prior to adoptive transfer, consistent with greater chemokine driven tumor trafficking and antigen recognition. The significant loss of CAR and Pmel T cells in the tumor, and the absence of a block in proliferation or mislocalization, suggested that VSV promoted apoptosis of the adoptively transferred cells via a mechanism that was T cell intrinsic.

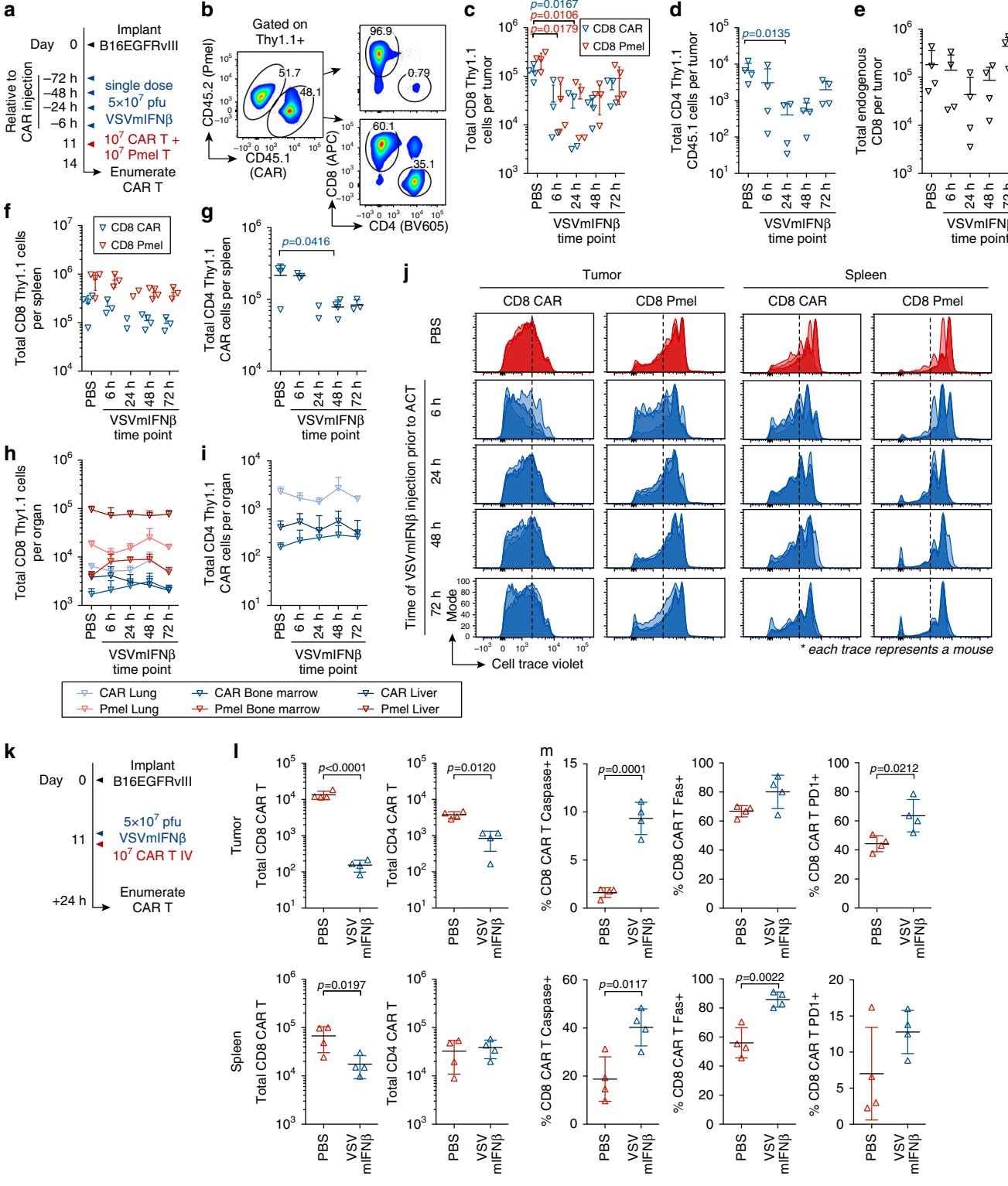

VSVmIFNβ-associated attrition was also evaluated in a subcutaneous CT2AEGFRvIII tumor model which supports VSV infection and transgene expression (Supplementary Fig. 3A-I). As observed in the B16EGFRvIII tumors, when CT2AEGFRvIII tumors were intratumorally treated with VSVmIFNβ 6 or 24 h prior to adoptive transfer, significantly fewer CD8 CAR T cells were recovered compared to a PBS injection. A trend toward CD4 CAR T cell attrition was also observed in the tumor, although it did not reach statistical significance. Cell trace violet dilution also did not reveal impaired proliferation as a mechanism which would

account for the reduction in CAR T cell number (Supplementary Fig. 3J, K).

Intratumoral VSVmIFNβ injection 6 h prior to adoptive transfer of CAR T cells promoted detectable attrition in the tumor and spleen as early as 24 h following administration (Fig. 2k, l). The loss of CAR T cells was more profound in the tumor than the spleen. Based on these data, it is not possible to distinguish whether the levels of CAR T cells in the spleen reflected recirculated cells from the tumor, or whether some virus infection- associated apoptosis directly occurred in the spleen.

**Fig. 2 Oncolytic VSVmIFNβ infection promotes T cell attrition in the tumor and spleen. a** Mice bearing B16EGFRvIII tumors received a single intratumoral injection of $5 \times 10^7$ pfu of VSVmIFNβ or PBS at various time points (6 to 72 h) prior to adoptive transfer of $1 \times 10^7$ activated Pmel T cells $+ 1 \times 10^7$ EGFRvIII CAR T cells on day 11. **b** Both the CAR retrovirus and the Pmel mice express the congenic marker Thy1.1. To distinguish CAR T cells from C57 WT mice and Pmel T cells, CAR T cells were prepared from CD45.1 donor mice. Three days post-adoptive transfer, the number of viable Thy1.1$^+$ CD8 (**C**) and CD4 (**d**) and endogenous CD8 (**e**) T cells were quantified in the tumor and represented as mean ± SD for $n = 4$/group. The number of viable Thy1.1 CD8 (**f**) and CD4 (**g**) T cells were quantified in the spleen and represented as mean ± SD for $n = 4$/group except in the VSV 6 h and 72 h time points $n = 3$ and VSV 24 h time point $n = 2$. The number of viable Thy1.1 CD8 (**h**) and CD4 (**i**) T cells were quantified in the lungs, liver, and left tibia and femur (bone marrow) and represented as mean ± SD for $n = 4$/group. **j** Pmel and CAR T cells were labeled with Cell trace violet and proliferation is shown in the tumor, and spleen. The Cell trace violet dilution for all four mice per group is overlaid. Data (**c-j**) are representative of two independent experiments. **k** B16EGFRvIII tumor bearing mice received PBS or $5 \times 10^7$ pfu of VSVmIFNβ intratumorally 6 h prior to adoptive transfer, and 24 h later, tumors and spleens were harvested. Total live CD8 and CD4 CAR T cells in the tumor (upper panels) and spleen (lower panels) is shown as mean ± SD for $n = 4$/group in (**l**). The percentage of live CD8 CAR T cells expressing cleaved caspase, Fas and PD1 in the tumor (upper panels) and spleen (lower panels) is shown as mean ± SD for $n = 4$/group in (**m**). The experiment in k–m was performed once. Symbols from (**h**) and (**i**) represent mean group data; symbol from all other panels represent individual mice. P-values were determined using a two-way ANOVA with a Tukey multiple comparisons post-test (**c**), a one-way ANOVA with a Tukey multiple comparisons post-test (**d, g**) or an unpaired two tailed t-test (**l**) on log-transformed data. P-values were determined using an unpaired two tailed t-test (**m**). Statistical significance set at $p < 0.05$, ns > 0.05. Source data are provided in the Source Data File.

The magnitude of the depletion was also greater than that observed in Fig. 2c, consistent with the enhanced proliferation which masked the early effect. Concomitant with this loss of CAR T cells, we observed a higher frequency of CD8 CAR T cells which expressed cleaved caspase, Fas, and PD1, in both the tumor and spleen, in VSVmIFNβ infected animals (Fig. 2m). Increased activation of CD8 and CD4 CAR T cells was also observed in the spleen 24 h after adoptive transfer into VSVmIFNβ infected mice (Supplementary Fig. 4).

We next sought to determine whether another OV would promote a comparable depletion of CAR T cells, or whether the effect was restricted to VSV. Analogous to Fig. 2a, we administered a single intratumoral dose of Reovirus between 24 and 72 h prior to CAR T delivery and quantified cells in the tumor, spleen, and draining inguinal lymph node 3 days later (Supplementary Fig. 5A). Fewer CD8 and CD4 CAR T cells were detected in the tumor if Reovirus was administered 24–72 h prior to adoptive transfer (Supplementary Fig. 5B, C). However, significant attrition was not observed in the spleen or draining lymph node (Supplementary Fig. 5D, E). A contraction of the endogenous CD8 compartment in the tumor was not observed (Supplementary Fig. 5F). Similar to the VSV-induced proliferative effect, we observed enhanced cell trace violet dilution in CAR T cells in the tumor and draining lymph node in Reovirus infected animals (Supplementary Fig. 5G). Both CD8 and CD4 CAR T cells were also more highly activated in the tumor, spleen, and tumor draining lymph node in Reovirus infected animals (Supplementary Fig. 5H). These data suggest that the attrition was common to both OVs, although the virally induced mechanism was more robust with VSVmIFNβ.

**Type I IFN is deleterious to CAR T cells**. Type I IFNs are pleiotropic regulators with highly context dependent, and sometimes opposing, functions including both the promotion and inhibition of proliferation and apoptosis[22]. Memory CD8 T cells have previously been shown to undergo type I IFN-induced apoptosis that was associated with Bim and Fas following poly IC or LCMV exposure[23–25]. As the most profound loss of CAR T cells occurred at the time points where we observed the peak concentration of IFNβ in the tumor (Figs. 1c, 2c), we next evaluated whether the attrition was related to the virus induced or transgene-driven type I IFN. We administered VSVmIFNβ or VSVGFP 6 h prior to CAR T cell transfer and 18 h later quantified viable CAR T cells in the tumor and spleen (Fig. 3a). We observed an attenuated attrition of CAR T cells in the tumor and no significant reduction in CAR T cell numbers in the spleen if VSVGFP was administered rather than VSVmIFNβ (Fig. 3b, c). The CAR T cell loss was correlated with the magnitude of type I

IFN induction in the tumor (Fig. 3d). Although VSV GFP induced a robust increase in the concentration of IFNβ in the tumor relative to PBS, it was significantly lower than VSVmIFNβ.

Since our findings suggested that a virus which generated reduced levels of type I IFN may be more amenable to combination therapy with CAR T cells, we evaluated the combination of VSVGFP with EGFRvIII CAR T cells in both non-irradiated and lymphodepleted animals (Fig. 3e). As with VSVmIFNβ, each modality conferred some tumor control, and the combination exerted minimal additional therapeutic benefit (Fig. 3f). Although VSVGFP induced less IFNβ than VSVmIFNβ, a modest attrition was nonetheless observed (Fig. 3b). We therefore reasoned that CAR T cells generated from donor mice which do not express the type I IFN receptor (IFNAR1) would be protected from the deleterious effects induced by VSVmIFNβ, regardless of the level of type IFN in the tumor (Fig. 3g). A 1:1 mixture of CAR T cells prepared from transgenic IFNAR KO (CD45.2) and WT (CD45.1) donor mice was injected into non-irradiated or lymphodepleted mice that received PBS or VSVmIFNβ IT. In the spleens of PBS injected mice, the ratio of IFNAR KO: WT CD8 CAR T cells was near or below 1(Fig. 3h, i). VSVmIFNβ administration provided a selective advantage for IFNAR KO CD8 CAR T over WT CD8 CAR T cells in both the spleen and tumor, and the magnitude of the enrichment was enhanced in lymphodepleted animals. Although IFNAR KO CD8 CAR T cells were less sensitive to the deleterious effects of VSVmIFNβ, the absence of the receptor did not provide complete protection in non-preconditioned animals, and more limited attrition was observed in the IFNAR KO compartment compared to the WT compartment in the tumor. In lymphodepleted mice, there was no statistical difference between the number of recovered IFNAR KO CAR T cells from PBS or VSVmIFNβ treated mice (Fig. 3j). IFNAR1 KO CAR T cells were not more permissive to VSV infection, nor did they undergo more apoptosis when infected in vitro (Supplementary Fig. 1C). As it has been previously shown that type I IFN signaling is necessary to protect CD8 T cells against NK attack in the context of virus infection[26,27], we hypothesized that the enhanced selection and recovery of IFNAR KO CD8 CAR T cells in the VSV-pretreated lymphodepleted animals may reflect the depletion of NK cells. Anti-NK1.1 antibody administration functionally reproduced the effect of TBI where the selective recovery of IFNAR KO CAR T cells was enhanced in the absence of NK cells (Supplementary Fig. 6). Therefore, IFNAR KO CAR T cells are protected against the direct apoptosis induced by VSV-derived type I IFN; however, subsequently rendered sensitive to NK attack in non-lymphodepleted animals.

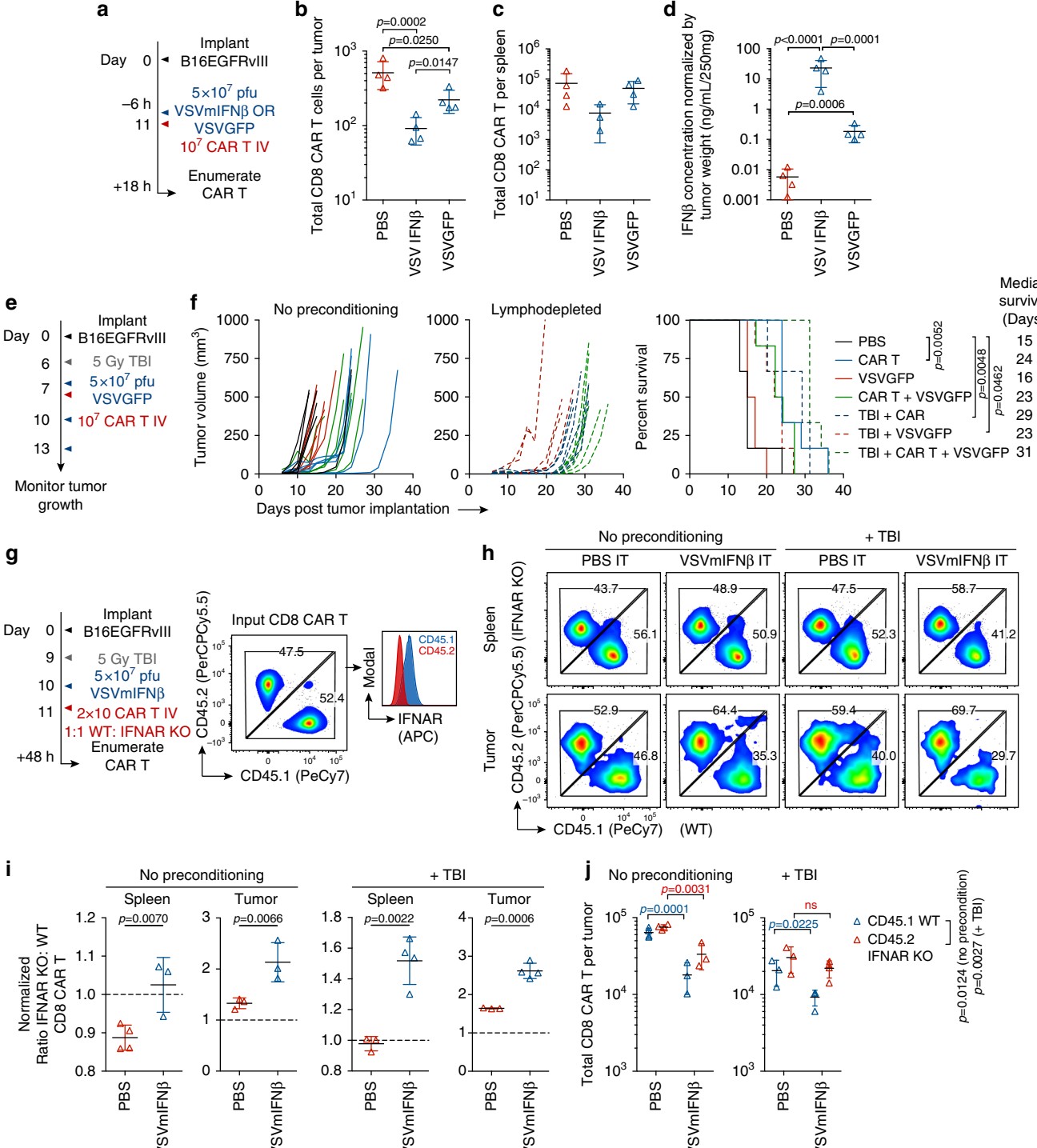

**Fig. 3 Oncolytic virus-associated type I IFN is deleterious to adoptively transferred CAR T cells. a** Mice bearing B16EGFRvIII tumors received a single intratumoral injection of $5 \times 10^7$ pfu of VSVmIFNβ or VSVGFP or PBS 6 h prior to adoptive transfer of $1 \times 10^7$ EGFRvIII CAR T cells on day 11. The next day, the number of viable CD8 CAR T cells was quantified in the tumor (**b**) and spleen (**c**) and represented as mean ± SD for $n = 4$/group. **d** IFNβ was quantified in the tumor homogenate, normalized by tumor weight, and represented as mean ± SD for $n = 4$/group. **e** Mice bearing B16EGFRvIII tumors were treated with $5 \times 10^7$ pfu VSVGFP or PBS 6 h prior to administration of $1 \times 10^7$ EGFRvIII CAR T cells on day 7. Select groups received a lymphodepleting dose of radiation (5 Gy TBI) on day 6, and 2 additional doses of $5 \times 10^7$ pfu VSVmIFNβ on days 10 and 13. $n = 6$/group. **f** Tumor growth is shown in the left panel and overall survival is shown in right panel. **g** Mice bearing B16EGFRvIII tumors received a single intratumoral injection of $5 \times 10^7$ pfu of VSVmIFNβ or PBS 48 h prior to adoptive transfer of $1 \times 10^7$ WT CD45.1 EGFRvIII CAR T cells + $1 \times 10^7$ transgenic IFNAR1 KO CD45.2 EGFRvIII CAR T cells. **h** The ratio of IFNAR KO to WT CD8 CAR T cells in the tumor and in the spleen is shown for a single representative mouse per group. **i** The IFNAR KO: WT ratio normalized to the input ratio and represented as mean ± SD for $n = 3$ or 4/group. **j** The total number of CD8 CAR T cells of each type was enumerated in the tumor and spleen and represented as mean ± SD for $n = 3$ or 4/group. Experiments were performed once. $P$-values were determined using a one-way ANOVA with a Tukey multiple comparisons post-test using log-transformed data (**b**, **d**), a Log-rank Mantel–Cox test (**f**), unpaired two tailed T tests (**i**) or a two-way ANOVA with a Sidak multiple comparisons post-test (**j**). Statistical significance set at $p < 0.05$, ns > 0.05. Source data are provided in the Source Data File.

**IFNβ elicits T cell intrinsic and CAR T specific effects**. To further evaluate the effect of type I IFN on CAR T cells, we utilized in vitro culture systems of CAR T cells alone, or cocultured with target B16EGFRvIII tumor cells in the presence or absence of recombinant IFNβ spanning the concentration range observed in Fig. 3d. Consistent with the in vivo loss of CAR T cells in VSVmIFNβ infected tumors, we observed an IFN dependent increase in the proportion of Annexin V positive apoptotic cells (Fig. 4a). The type I IFN receptor signals through the Janus kinase (JAK)–signal transducer and activator of transcription (STAT) pathway, and inhibition of signal transduction with the JAK 1/2 inhibitor Ruxolitinib prevented the IFNβ induced increase in apoptosis. This effect was antigen independent, as it was observed both in the presence and absence of EGFRvIII (Fig. 4a, b). Similar to the results in vivo (Fig. 2m), we observed a dose dependent upregulation of Fas and cleaved caspase (Fig. 4c, d). Although both CAR T cells and untransduced T cells were sensitized to apoptosis by IFNβ, higher levels of Annexin V and cleaved caspase were observed in the CAR T cells, suggesting both a T cell intrinsic IFN mechanism as well as a CAR T specific effect. In particular, the CAR T cells which underwent IFN-induced apoptosis expressed the highest levels of the CAR on the surface (Fig. 4e).

High level expression of the CAR molecule on T cells has previously been shown to sensitize them to tonic signaling, and to promote exhaustion and apoptosis[28,29]. Indeed, we observed an IFNβ dose- dependent increase in CAR expression on the surface of transduced CD8 T cells that could be blocked by JAK1 inhibition (Fig. 4f, g). CD8 CAR T cells exhibited a higher baseline level of activation than control untransduced CD8 T cells in the absence of the target, consistent with tonic signaling through the CAR, and greater than 90% of CD8 CAR T cells expressed either CD25, CD69, or both, in the presence of 10 ng/mL IFNβ (Fig. 4h, i). Although the UTD cells also became activated in the presence of IFNβ, the effect was markedly attenuated compared to the CAR T cells. Furthermore, the cells with the highest-level expression of the CAR were also the most activated (Fig. 4j).

In parallel, we observed a dose dependent increase in the expression of the inhibitory receptors LAG3, PD1, and TIM3 that was T cell intrinsic. However, this effect was also markedly enhanced in the transduced cells and correlated with the expression level of the CAR (Fig. 4k, l). As was consistent with the regulation of the CAR and the activation state, the upregulation of inhibitory receptors was greatest between 0 and 1 ng/mL, and plateaued at higher doses. To further confirm the effect of type I IFN on the upregulation of the CAR and subsequent induction of inhibitory receptor expression, we employed a coculture system of CAR T cells with either mock infected or VSVmIFNβ infected B16EGFRvIII target tumor cells (Fig. 4m, n). While CAR expression was enhanced on WT CD8 T cells cocultured with infected tumor cells in an MOI dependent manner, we observed no change in the surface expression of the CAR on the IFNAR1 KO CD8 T cells (Fig. 4m). Furthermore, WT CD8 CAR T cells exhibited an elevated inhibitory receptor profile in the presence of VSVmIFNβ infected tumor cells that was not observed in the IFNAR1 KO CD8 CAR T cells (Fig. 4n).

We observed more modest induction of apoptosis, activation, CAR dysregulation, and inhibitory receptor upregulation in the CD4 CAR T cell compartment in response to IFN (Supplementary Fig. 7A-H). The difference between the CD4 and CD8 CAR T cell compartments possibly derives from the higher level of CAR expressed on the CD8 cells (Supplementary Fig. 7E), or the higher level of the IFNAR1 on the CD8 CAR T cells (Supplementary Fig. 7I), although intrinsic differences in the way that CD4 and CD8 T cells respond to type I IFN signaling cannot be ruled out.

We performed the same in vitro IFNβ treatments to a third generation murine CD19 CAR in which the 139 EGFRvIII-specific scFv was replaced with the FMC63 scFv, an scFv previously reported to have low aggregation propensity and thus to undergo minimal levels of tonic signaling[28]. As with the EGFRvIII CAR, IFNβ promoted higher level expression of the CAR molecule on the surface of the T cells and supported an activated phenotype (Supplementary Fig. 8A–C). In contrast, however, to the EGFRvIII CAR, the higher level expression of the CD19 specific CAR was not associated with increased levels of inhibitory receptor expression over and above the T cell intrinsic IFNβ mediated changes observed in the untransduced cells (Supplementary Fig. 8D).

In order to validate the murine findings in human cells, EGFRvIII CAR T cells were generated from healthy donor PBMCs using a retroviral construct encoding the same targeting scFv and the corresponding human signaling domains[30]. Analogous to Fig. 4, untransduced or CAR T cells from three independent donors were cultured in vitro in the presence of recombinant human IFNβ for 48 h. We observed a dose dependent increase in apoptosis as measured by cleaved caspase 3 staining in both CD4 and CD8 CAR transduced and untransduced T cells (Fig. 5a). Consistent with the mouse studies as well, we also observed elevated expression of the inhibitory receptors PD1, TIM3, and in particular, LAG3 in the presence of increasing concentrations of IFNβ (Fig. 5b–e).

**Alternative scheduling may facilitate combination therapy**. Since the administration of VSV and CAR T cells in short succession was limiting T cell engraftment and causing therapeutic interference, we explored alternative scheduling. We administered CAR T cells to lymphodepleted animals and waited 5 days before administering the first dose of VSV (Supplementary Fig. 9A). Animals received three doses of VSVmIFNβ followed by either another 3 doses of VSVmIFNβ or VSVGFP when tumors were palpable. Both the treatment of smaller tumors with VSVmIFNβ on days 9–14 and the administration of VSVGFP to palpable tumors on days 23–28 was chosen to minimize the amount of IFNβ in the tumor. Although CAR T cell attrition in the blood on day 18 was minimized, 6 doses of VSVmIFNβ provided little therapeutic benefit over the CAR T cells alone. Administration of VSVGFP on days 23, 25, and 28 provided a very modest extension in survival over CAR T cells. Since virus-associated type I IFN upregulates inhibitory receptors on CAR T cells (Fig. 4k, n) as well as PD-L1 and Galectin 9 on tumor cells (Supplementary Fig. 9B), we reevaluated the therapeutic potential of CAR T cells with VSVGFP in combination with a cocktail of anti-PD1, anti-TIM3, and anti-LAG3. A modest, but significant extension of survival was observed in the animals treated with CAR T cells, VSVGFP and checkpoint inhibitors compared with CAR T cells alone. A single animal in this group remained tumor free until the end of the experiment (day 136) (Supplementary Fig. 9C). Therefore, reducing the amount of type I IFN in the tumor produced by the virus, employing checkpoint inhibitors to counteract the residual IFN-induced effects on the CAR T cells, and modifying treatment scheduling, may enable combination therapy.

**IFNAR KO CAR T cells enable combination therapy**. Our data suggested that CAR T cells which are refractory to type I IFN would be more readily amenable to combination therapy with OVs. In order to make this approach clinically translatable, we explored CRISPR Cas9 genetic disruption of *IFNAR1* in CAR T cells. Nucleofection of two *IFNAR1* targeted crRNA RNP complexes on the day following retroviral CAR transduction ablated IFNAR1

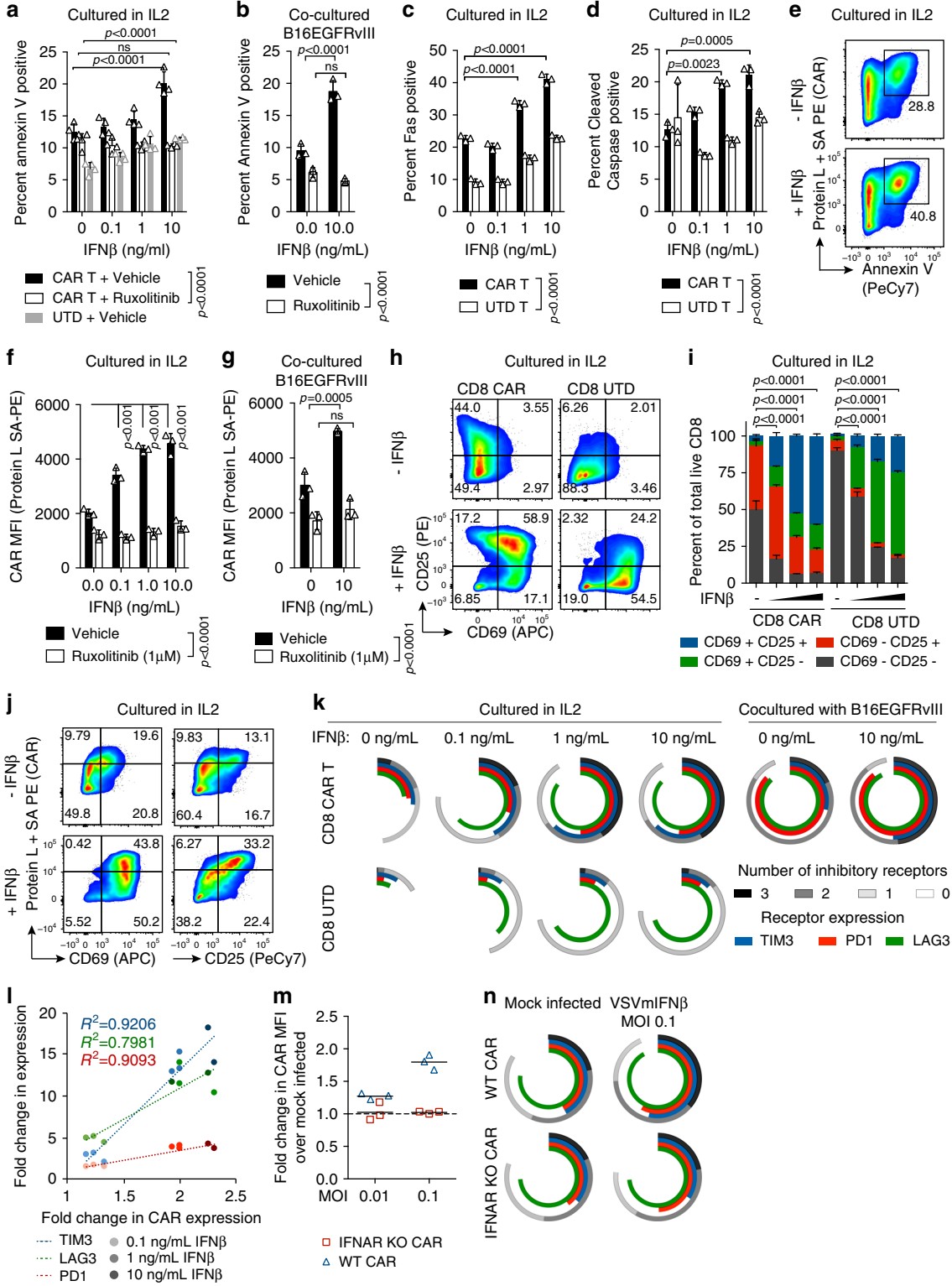

expression and generated CAR+ IFNAR1− CD8 and CD4 populations with approximately 92 and 85% efficiency, respectively (Fig. 6a). CRISPR modified CAR T cells were functionally insensitive to the deleterious effects of recombinant IFNβ, and did not upregulate the CAR, Fas, or inhibitory receptors (Fig. 6b–d).

As IFNAR1 KO CAR T cells were at least partially resistant to the VSVmIFNβ induced attrition in vivo, and were also resistant to CAR expression dysregulation in vitro, we hypothesized that these type I IFN insensitive transgenic CAR T cells would provide

superior tumor control in combination with VSVmIFNβ over CAR T cells generated from WT donor mice. We evaluated tumor growth and overall survival in both non-preconditioned and lymphodepleted mice that received IFNAR KO EGFRvIII CAR T cell monotherapy or combination therapy with VSVmIFNβ (Fig. 6e). For the first time in our combination studies with VSV administered 6 h prior to CAR T cells, we observed a reduced tumor growth rate and a significant extension of survival in lymphodepleted mice which received both IFNAR1

**Fig. 4 Recombinant IFNβ promotes apoptosis, activation, inhibitory receptor expression, and dysregulated expression of the CAR. a** Annexin V expression on CD8 CAR T cells or untransduced (UTD) T cells cultured in IL2 (50 U/mL) in the presence or absence of recombinant murine IFNβ and 1μM Ruxolitinib (JAK1/2 inhibitor) for 48 h. **b** Annexin V expression on CD8 CAR T cells cocultured with B16EGFRvIII cells at an E:T ratio of 1:5 in the presence or absence of recombinant murine IFNβ and/or 1μM Ruxolitinib. Fas (**c**) or cleaved caspase (**d**) expression on CD8 CAR T cells or UTD cells cultured in IL2 with IFNβ. **e** Median fluorescence intensity (MFI) of CAR expression measured using Biotin-Protein L and streptavidin- PE (Protein L + SA PE) compared to Annexin V on CAR T cells cultured in IL2 with or without IFNβ (10 ng/mL). CAR expression on CD8 CAR T cells cultured in IFNβ for 48 h in the absence (**f**) or presence of target tumor cells (**g**). Representative (**h**) and mean (**i**) expression of CD25 and CD69 for CAR T cells cultured in IL2 with IFNβ. Statistical comparisons are indicated for CD25⁻ CD69⁻ populations in (**i**). **j** Co-expression of the CAR and activation markers on CD8 CAR T cells. **k** Mean expression of PD1, LAG3, TIM3 on CD8 CAR T or UTD cells cultured in IL2 or in the presence of B16EGFRvIII cells (E:T 1:5). **l** Fold change in inhibitory receptor and CAR expression is shown for cells grown in IL2 with additional recombinant IFNβ relative to culture in IL2 alone. **m** WT or transgenic IFNAR1 KO CAR T cells were cocultured with B16EGFRvIII (E:T ratio of 1:5) which were mock infected or infected with VSVmIFNβ 6 h prior to coculture. Surface CAR expression is represented as fold change relative to coculture with mock infected tumor cells. **n** Inhibitory receptor expression was quantified on WT or IFNAR1 KO CD8 CAR T cultured as in (**m**). Data are representative of two independent experiments (panels **a-l**). Experiments in panels (**m**, **n**) were performed once. Means ± SD of n = 4 (panel **a**) or n = 3 (panels **b-d**, **f**, **g**, **i**, **k**, **m**, **n**) technical replicates are shown. P-values were determined using a two-way ANOVA with a Tukey multiple comparisons post-test (**a-d**, **f**, **g**, **i**). Statistical significance set at p < 0.05, ns > 0.05. Source data are provided in the Source Data File.

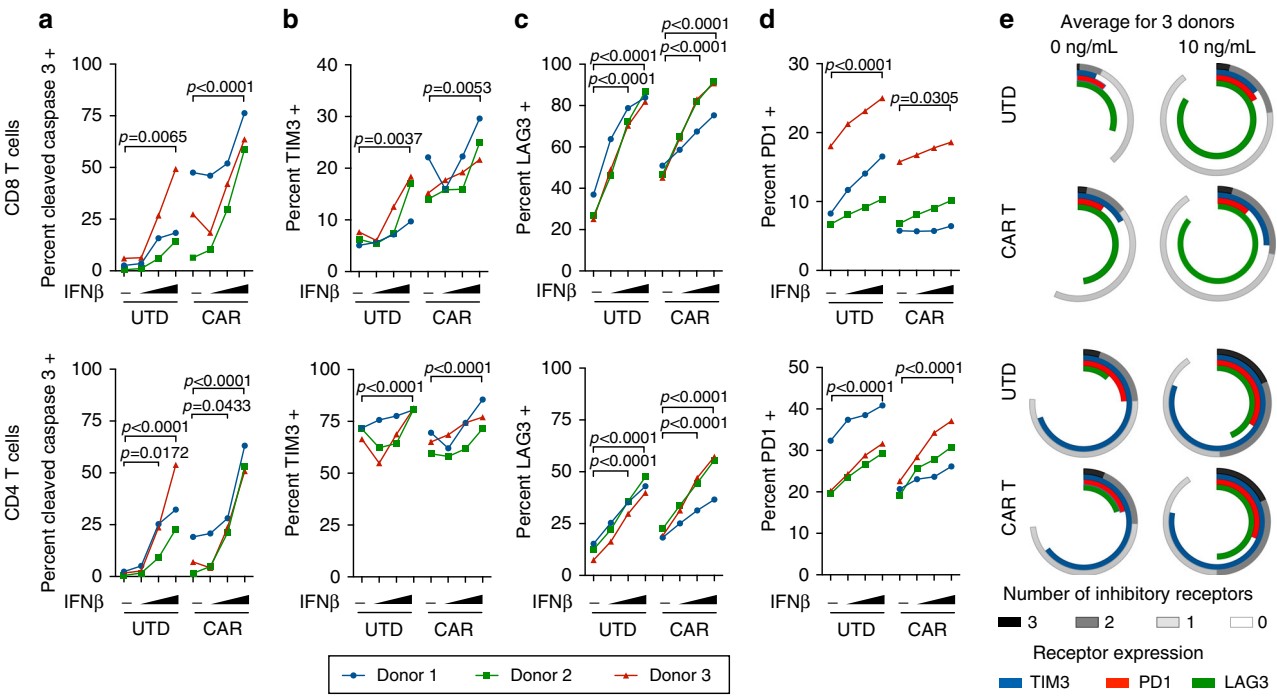

**Fig. 5 Recombinant IFNβ promotes human CAR T cell apoptosis and inhibitory receptor expression.** CAR and untransduced (UTD) T cells from three healthy donors were cultured in the absence or presence of human IFNβ (0, 0.1, 1, or 10 ng/mL) for 48 h and analyzed for the expression of (**a**) cleaved caspase 3, (**b**) TIM3, (**c**) LAG3, and (**d**) PD1. **e** Average expression of individual and combinations of inhibitory receptors for three donors. Data are representative of two independent experiments. Data points indicate the average of two technical replicates for each donor and P-values were determined for all three donors using a two-way repeated measures ANOVA with a Tukey multiple comparisons post-test. Source data are provided in the Source Data File.

KO CAR T cells and three doses of VSVmIFNβ compared to those animals which only received CAR T cells (Fig. 6f). In the non-lymphodepleted setting, tumor outgrowth was similar in animals receiving CAR T cells with or without VSVmIFNβ. We further confirmed that in lymphodepleted animals, IFNAR KO and WT CAR T cells provided similar tumor control in combination with PBS injections; however, IFNAR KO CAR T cells provided a modest but significant survival advantage over WT CAR T cells when combined with VSVmIFNβ (Fig. 6g, h).

## Discussion

It has been widely assumed that OVs can convert immunological desert-like (cold) tumors into immunologically inflamed

(hot) tumors, resulting in higher rates of response and regression. However, our data here show that the immunological heat generated by tumor infection by an OVs is a complex multi-factorial process which is not automatically beneficial for immunotherapy. Our initial hypothesis was that OV-derived inflammation would prime optimal conditions for CAR T cell infiltration and functionality, and thus would work synergistically with virus-induced oncolysis. Consistent with this overall hypothesis, tumor infection by VSV generated a chemokine profile rich in CXCL10 and CCL5 which we predicted would be highly favorable to the recruitment of adoptively transferred CAR T cells. Despite the enhanced chemokine profile, we observed profound attrition of CAR T cells in VSV and Reovirus-infected tumors. The magnitude of the CAR T cell loss

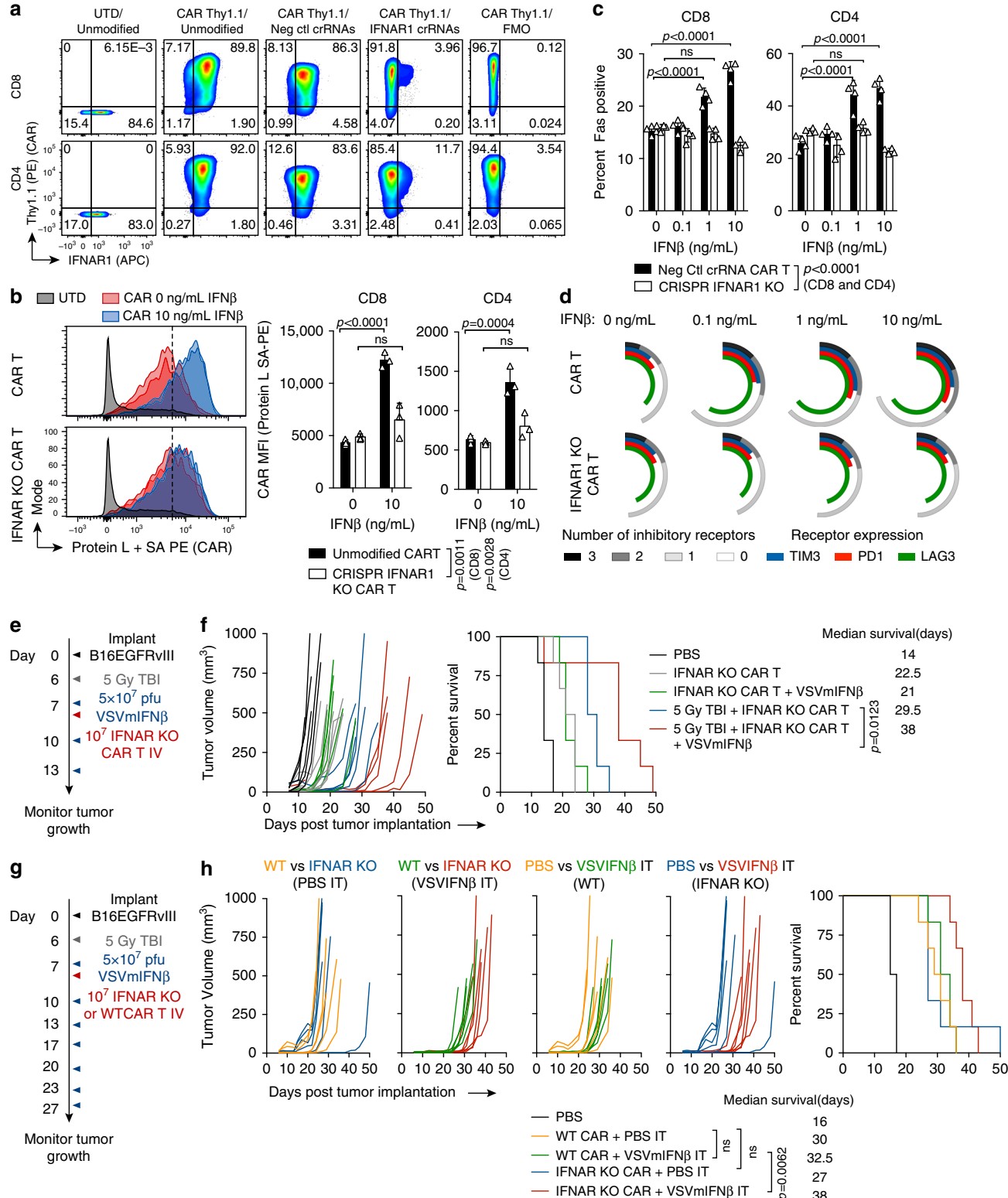

was proportional to the concentration of type I IFN in the tumor, and CAR T cells which did not express the IFNAR1 were largely refractory to virus-induced attrition in the setting of lymphodepletion or NK cell depletion. This effect was T cell intrinsic as fewer CAR T cells and Pmel T cells were recovered from virus infected tumors. However, unique aspects of CAR T biology were revealed through in vitro culture in recombinant IFNβ. Both in vivo and in vitro, CAR T cells which were exposed to type I IFN were sensitized to apoptosis and displayed greater activation marker and inhibitory receptor expression. Furthermore, IFNβ-induced T cell apoptosis and inhibitory receptor expression were confirmed to also occur in human CAR and untransduced T cells. Based on these findings, we further demonstrated enhanced tumor control and overall survival in mice treated with IFNAR1 KO CAR T cells in combination with VSVmIFNβ. These findings identify IFNAR1 as one of likely

**Fig. 6 Type I IFN resistant CAR T cells provide enhanced therapy with VSVmIFNβ in lymphodepleted mice. a** CAR T cells were genetically modified using CRISPR Cas9 one day after transduction by nucleofection of an RNP complex consisting of Cas9 duplexed with tracrRNA and two *IFNAR1* specific or two negative control crRNAs. 48 h following modification, expression of the CAR (Thy1.1) and the IFNAR1 is shown. **b** Two days after modification, CAR T cells were cultured in IL2 (50 U/mL) in the absence or presence of additional recombinant mouse IFNβ. CAR expression is shown for representative CD8 CAR T cells (left) and quantified in three replicates in CD8 and CD4 CAR T cells (right). **c** The percent of CRISPR IFNAR1 KO or control CD8 and CD4 CAR T cells expressing Fas is shown. **d** Inhibitory receptor expression (PD1, LAG3, TIM3) quantified on CRISPR IFNAR1 KO or control CD8 CAR T cells cultured in IL2 in the absence or presence of additional IFNβ. Data shown are representative of two independent experiments. Technical replicates are shown ± SD ($n = 3$ (**b**) $n = 4$ (**c**)). **e** Mice bearing B16EGFRvIII tumors were treated with $5 \times 10^7$ pfu VSVmIFNβ or PBS 6 h prior to administration of $1 \times 10^7$ IFNAR1 KO EGFRvIII CAR T cells on day 7. Select groups received two additional doses of $5 \times 10^7$ pfu VSVmIFNβ on days 10 and 13. Select groups received a lymphodepleting dose of radiation (5 Gy TBI) on day 6. $n = 6$/group. **f** Tumor growth is shown in the left panel and overall survival is shown in right panel. **g** Mice bearing B16EGFRvIII tumors received a lymphodepleting dose of radiation (5 Gy TBI) on day 6 and were treated with $5 \times 10^7$ pfu VSVmIFNβ or PBS 6 h prior to administration of $1 \times 10^7$ WT or IFNAR1 KO EGFRvIII CAR T cells on day 7. Select groups received six additional doses of $5 \times 10^7$ pfu VSVmIFNβ on days 10,13,17,20,23, and 27. $n = 6$/group. Experiments in (**e**) and (**g**) were performed once. *P*-values were determined using the Log-rank Mantel-Cox test (**f**, **h**) and a two-way ANOVA with a Tukey multiple comparisons post-test (**b**, **c**). Statistical significance set at $p < 0.05$, ns > 0.05. Source data are provided in the Source Data File.

many genetic engineering strategies to CAR T cells which will improve combination with highly inflammatory adjuvants.

Rapidly induced innate responses to pathogens, such as OVs, are perfectly suited to the priming, activation and propagation of adaptive immune responses to that pathogen. This occurs through a carefully regulated sequence of inflammation induction and subsequent active resolution, followed by a shift of chemokine and cytokine secretion towards a T cell tropic profile. Whilst this response is highly appropriate in the initiation of an adaptive immune response to an infection, and in theory against an infected tumor, our data here show that the type I IFN response to viral infection can act deleteriously against a pre-existing effector T cell response in the form of adoptively transferred CAR T cells. IFNα/β have been shown to have highly context dependent effects on T cell biology. They both act as signal 3 cytokines during the priming phase[19,31], yet promote apoptosis of CD44[hi] memory T cells early during infection[22–24]. Consistent with our data that high-level type I IFN-driven immunological heat may be contextually inappropriate, intratumoral injection of the TLR 3 and 9 ligands Poly(I:C) and CpG in combination with adoptive transfer of Pmel T cells has previously been associated with a modest increase in survival, coupled with the detection of significantly fewer Pmel T cells in the tumor and lymph nodes[32]. While high level IFN may facilitate the priming of tumor reactive T cells during oncolysis, it may also promote memory T cell attrition to make room for more diverse and effective T cell responses to novel pathogens[17]. Although multiple preclinical and clinical reports have documented increased TIL density following virus treatment, we reason that several factors are necessary to interpret how an OV and its associated inflammation may dynamically potentiate the number of T cells in a tumor including: (1) the time point of T cell enumeration relative to virus injection, (2) the dose and route of administration, (3) the differentiation state of the T cells at the time of virus exposure, and (4) the specificity of the T cells.

Of particular importance to our model, type I IFN ligands bind to species-specific IFNAR1/2[33] with limited cross-reactivity. Mouse and human IFNAR1 and 2 have an overall sequence homology of 50.7 and 50.4%, respectively[34], and residues in both IFNβ and IFNAR1 which are key to the interface are variably conserved[35]. Therefore, the preclinical testing of such combination approaches in immunocompetent models is of the utmost importance.

The role of type I IFN is highly complex in this model. The tumor selectivity of VSVmIFNβ derives in large part from the expression of the transgene, and more broadly, type I IFN insensitivity of tumor cells underlies the tumor tropism of most oncolytic vectors. For other OVs which do not encode IFN, it is now clear that even if tumor cells lack IFN sensitivity, multiple stromal elements of tumors are able to induce type I IFN during infection. Our data suggest that both the choice of OV and its genetic modifications may render it more or less amenable to combinations with CAR T cells. Intrinsic properties such as the inflammatory kinetics of the infection and amount of type I IFN produced are key determinants of successful combination approaches. Indeed, we observed variable degrees of CAR T cell depletion with VSVmIFNβ, VSVGFP, and Reovirus. Many viruses have evolved mechanisms that limit or quench type I IFN signaling. In particular, the vaccinia virus B18R gene serves as soluble type I IFN decoy receptor[36], therefore highlighting the potential of vaccinia virus or heterologous viruses engineered to encode B18R[37] for CAR T combination. Indeed, the expression of the B18R gene may have facilitated the combination of oncolytic vaccinia virus with murine mesothelin CAR T cells in a TC1-mesothelin model[38]. Although we have described a mechanism in which type IFN plays a major role in the therapeutic interference that we observed with VSV, it is certainly possible that there are other pathogen- or danger-associated molecular pattern molecules which are released upon oncolysis which could adversely affect CAR T cell health.

The CAR T cell attrition associated with VSVGFP was attenuated compared to VSVmIFNβ; however, a degree of attrition was nonetheless detected, and combination therapy was not enhanced. We reason that genetic modifications can be made to insulate CAR T cells and enable them to be effective in inflamed microenvironments. IFNAR1 KO CAR T cells were partially resistant to the VSVmIFNβ induced attrition in vivo and provided improved tumor control in a combinatorial setting. In vitro, the apoptosis associated with recombinant IFNβ was not of a sufficient level to account for the degree of attrition observed in vivo. Therefore, although type I IFN insensitivity is an important factor, additional modifications to the CAR T cells are likely to further improve combination therapy. We have observed higher levels of both Fas and PD1 on CAR T cells in infected tumors, and the expression of their ligands can be regulated by viral infection. CRISPR knock out and retroviral introduction of dominant negative receptor approaches have been successfully employed in murine CAR T cells models[39,40]. Future studies will more comprehensively characterize secondary mediators of OV infection which may be deleterious to CAR T cell survival or function.

An aspect of virus-associated interference with CAR T therapy is T cell intrinsic, as we observed attrition of CAR T cells, adoptively transferred Pmel T cells, and the endogenous T cell compartment. However, in vitro, the outcome of type I IFN conditioning was exaggerated in CAR T cells compared to

untransduced T cells activated by the same protocol. The IFNβ induced activation, apoptosis, and inhibitory receptor expression occurred in an antigen independent manner and was correlated with high level expression of the CAR. We therefore hypothesize that through the upregulation of the CAR, IFNβ promoted tonic signaling. The clone FMC63 scFv specific to CD19 has been reported to have low aggregation and tonic signaling propensity[28], and consistent with this result, we observed an attenuated IFN profile in the CD19 CAR compared to the EGFRvIII CAR. Therefore, the scFv properties are likely to be an important determinant of how readily the CAR may be combined with inflammatory adjuvants.

It has been previously reported that the type I IFN gene signature is associated with enhanced functionality of a CAR configuration where 41BB-L co-stimulation is co-expressed rather than incorporated into the CAR molecule[41]. Not only has this particular signature specifically been reported for the 1928z-41BBL configuration and not the 1928BBz which more closely resembles our construct, the reported concentration of IFNβ was 2–3 orders of magnitude lower than that induced by VSV. Thus, we reason that the consequence of IFNβ exposure is not only dependent on antigen stimulation and is potentiated by cis/trans-co-stimulation, but also that it is context and concentration dependent, as is well known for type I IFNs.

In order to counteract the immunosuppressive TME, a therapeutic priority has been placed on the development of agents which inflame the tumor, particularly in the context of immune checkpoint blockade combination therapy[42–44]. However, immunologic heat associated with viruses or other adjuvants is counterbalanced by the strong engagement of physiologic feedback mechanisms meant to prevent immunopathology. We and others have observed the upregulation of inhibitory receptors on T cells, as well as ligands of these receptors on tumor and or stromal cells in virus infected tumors, in part, via an IFN dependent mechanism[45–49]. T cell intrinsic negative feedback can also limit highly activated tumor reactive T cells. In particular, treatment with anti-CTLA-4 and anti-PD-1 therapy in the low tumor burden state can lead to IFNγ dependent activation induced cell death of T cells[50]. The concentration of type I IFN induced in the tumor should be considered not only for adoptive T cell approaches, but also more broadly in the setting of repeat OV dosing where the endogenous T cell compartment may be depleted.

Our study herein demonstrates that inflammation induced by OVs needs to be understood as a multi-component phenomenon which can be either positive or negative for the generation of anti-tumor immunity. OV-associated type I IFN has negative consequences for CAR T cell viability, and rendering CAR T cells insensitive to type I IFN facilitates combination therapy. We emphasize that caution should be employed when using agents intended to inflame the tumor, and that such combination approaches are best evaluated in immunocompetent models in which there are no limitations associated with cross-reactivity between the CAR T cells and their environment.

## Methods
**Experimental design**. These experiments were designed to evaluate the potential benefits and shortcomings of OV and CAR T cell combination therapy, and further to develop strategies to overcome the barriers to their use. The investigators were not blinded to the allocation of groups during experiments or subsequently during the analysis. Although statistical methods were not used to predetermine sample size, sample sizes were chosen on the basis of estimates from pilot experiments and previously published results. Animals were randomized to treatment groups following tumor implantation. Pre-established exclusion criteria included T cell enumeration only in tumors weighing > 50 mg. Spleens with poor overall viability from Fig. 2f, g, d were not included in the analysis. The survival end point was reached when the tumor size reached 1 cm in diameter. The n values and particular

statistical methods are indicated in the figure legends as well as the statistical analysis section.

**Cell lines and viruses**. B16 murine melanoma cells, BHK, L929, and 293T cells were originally obtained from ATCC and maintained in DMEM (HyClone) + 10% FBS (Life Technologies). Cells were tested for mycoplasma using the MycoAlert Mycoplasma Detection Kit (Lonza). The B16EGFRvIII cell line was generated by retroviral transduction of B16 cells with the pBABE PURO vector encoding the murine EGFRvIII[51] modified by the deletion of 500 amino acids from the intracellular domain of the protein. A clonally derived cell line was subsequently maintained in 1.25 µg/mL of puromycin (Sigma). The CT2AEGFRvIII cell line[52] was maintained in DMEM + 10% FBS. The expression of EGFRvIII was verified by flow cytometry using the anti-human EGFRvIII antibody clone L8A4 (Absolute Antibody #Ab00184-1.1, dilution 1:100) and anti-mouse IgG1 (Biolegend #406608, clone RMG1-1, dilution 1:100). The PG13-139-CD8-CD28BBZ-F10 retroviral producer cell line was obtained from Dr. Steven Rosenberg and maintained in DMEM + 10% FBS[30].

VSV expressing murine IFNβ or GFP was rescued from the pXN2 cDNA plasmid[15,16] and propagated on BHK cells at low multiplicity of infection. 24 h post infection, supernatant was harvested, filtered through a 0.22 µm filter to remove debris and purified through a 10% sucrose cushion. Virus titers were determined by plaque assay on BHK cells. Wild-type Reovirus type 3 (Dearing strain) was obtained from Oncolytics Biotech (Calgary, AB, Canada) and stock titers were measured by plaque assay on L929 cells.

**Mice**. Female C57BL/6 (stock 000664) (CD45.2) and B6.SJL-Ptprca Pepcb/BoyJ (stock 002014) (CD45.1) mice were obtained from The Jackson Laboratory and female B6.129S2-Ifnar1tm1Agt/Mmjax (stock 32045⁻JAX) (IFNAR1 KO; CD45.2) mice were obtained from MMRC JAX. All mice were obtained at 6–8 weeks of age and maintained in a specific pathogen-free BSL2 biohazard facility. Pmel mice (originally obtained from The Jackson Laboratory (stock 005023); Thy1.1, CD45.2) were bred at the Mayo Clinic, and splenocytes from female mice were harvested between 8 and 14 weeks of age for adoptive transfer experiments. Experimental mice were co-housed and exposed to a 12:12 h light-dark cycle with unrestricted access to water and food. The ambient temperature was restricted to 68 to 79°F and the room humidity ranged from 30 to 70%. All animal studies were conducted in accordance with and approved by the Institutional Animal Care and Use Committee at Mayo Clinic.

**Murine CAR T cell preparation**. The EGFRvIII third generation MSGV1 retroviral CAR construct contains the CD28, 4-1BB, and CD3z moieties, in tandem with the scFv derived from the human monoclonal antibody 139, and the marker Thy1.1[21]. Alternatively, the EGFRvIII specific 139 scFv was replaced with the FMC63 CD19 specific scFv. CAR T cells were prepared according to previously reported protocols[53]. Briefly, splenocytes were isolated from donor C57BL/6 CD45.2 or CD45.1 or IFNAR1 KO mice, made into a single cell suspension, and cultured in RPMI (HyClone) supplemented with 10% FBS, 50 µM 2-Mercaptoethanol (Sigma), 1% PenStrep (Corning), 1% NEAA (Corning), 1% Sodium Pyruvate (Corning), 50 U/mL human IL2 (Novartis) and 2.5 µg/mL Concanavalin A (Sigma). Retroviral supernatant was produced from 293T cells co-transfected with the MSGV1 retroviral plasmid and the helper plasmid pCL Eco (Imgenex) and T cells were transduced on RetroNectin-coated plates (Takara) 2 days after stimulation. Cells were split one day after transduction and used for in vitro analysis or in vivo administration on day 4 or 5. Transduced cells were identified by the expression of Thy1.1 and a representative gating scheme identifying murine CAR T cells is shown in Supplementary Fig. 10.

**Human CAR T cell preparation**. Peripheral blood mononuclear cells (PBMCs) were isolated from healthy donor apheresis cones obtained through the Mayo Clinic Blood Donor Center[54] (approved by the Division of Transfusion Medicine Research Committee at Mayo Clinic and determined to be IRB exempt). Informed consent was obtained from all donors for the use of their sample for research purposes. Cells were isolated using Lympholyte-H density separation (Cedarlane) and cultured in AimV Media (ThermoFisher) supplemented with 5% human AB serum (Sigma) and 1% PenStrep and stimulated with 100 U/mL of human IL2 and 50 ng/mL anti-CD3 OKT3 antibody (Biolegend #317326). 48 and 72 h later, cells were transduced twice with retroviral supernatant collected from the PG13-139-CD8-CD28BBZ-F10 producer cell line on RetroNectin-coated plates (Takara). Cells were split every two days and collected 4 days after the second transduction for in vitro experiments. Transduced cells were identified using a tetramer composed of biotin-PepvIII and streptavidin-AF647 and a representative gating scheme identifying murine CAR T cells is shown in Supplementary Fig. 10. The PepvIII tetramer was generated by incubating PepvIII-biotin (Proteomics Core Mayo Clinic) with SA-AF647 (Invitrogen) at a 10:1 molar ratio and washed 4x using a 30 kDa MCWO filter.

**Pmel T cell preparation**. Splenocytes from Pmel mice were stimulated with the hgp100₂₅₋₃₃ peptide (2 µg/mL) in 50 U/mL IL2 and passaged analogously to CAR T cells. Cells were used on day 5 for in vivo administration.

**In vitro CAR cultures**. Mouse CAR T cells were cultured in human IL2 (50 U/mL) or at an E:T ratio of 1:5 with B16EGFRvIII cells for 2 days in the absence or presence of 0.1–10 ng/mL mouse recombinant IFNβ (R&D Systems). Additionally, 1 μM Ruxolitinib (InvivoGen) or vehicle (DMSO, Sigma) was added to block IFNAR1 signal transduction. Human CAR T cells were cultured in human IL2 (100 U/mL) for 2 days in the absence or presence of 0.1–10 ng/mL of human recombinant IFNβ (Peprotech).

**Cas9 RNP mediated editing of CAR T cells**. Cas9 RNP mediated editing of CAR T cells was carried out on day 3 of culture (one day after retroviral transduction) according to previously published methods[55]. Two IFNAR1 targeting crRNAs (Mm.Cas9.IFNAR1.1.AA (TCAGTTACACCATACGAATC) and Mm.Cas9. IFNAR1.1.AB (GCTTCTAAACGTACTTCTGG)) and Alt-R CRISPR-Cas9 Negative Control crRNAs #1 and #2 were ordered from Integrated DNA Technologies (IDT). Alt-R crRNA and Alt-R tracrRNA duplexes were prepared at equimolar concentrations and annealed at 95 °C for 5 min. 150 pmol of each duplex was precomplexed with 120 pmol TrueCut Cas9 Protein v2 (Thermo Fisher Scientific) for 10–20 min to modify $1 \times 10^7$ cells. T cells were mixed with RNP complex and Cas9 Electroporation Enhancer (4 uM, IDT). Nucleofection was performed using the Amaxa P4 Primary Cell kit and 4D-Nucleofecter (Lonza) using the program CM137.

**In vivo studies**. Mice were challenged subcutaneously with $3 \times 10^5$ B16EGFRvIII cells in 100 μL PBS (HyClone). Subcutaneous tumors were treated with up to 6 doses of $5 \times 10^7$ pfu of VSVmIFNβ or VSVGFP delivered intratumorally in 50 μL of PBS and a single dose of $10^7$ CAR T cells delivered intravenously by tail vein injection in 100 μL of PBS. Tumors were measured using calipers 3 times per week, and mice were euthanized using $CO_2$ when tumors reached 1.0 cm in diameter. For experiments requiring radiation, mice were subjected to 5 Gy TBI 24 h before CAR therapy. Mice received a checkpoint inhibitor cocktail of 100 μg each anti-mouse TIM3 (clone RMT-23 BioXCell #BE0115), anti-mouse PD1 (clone RMP1-14 BioXCell #BE0146), and anti-mouse LAG3 (clone C9B7W BioXCell #BE0174) per dose intraperitoneally (IP). Control mice received 300 μg of control rat IgG (Jackson ImmunoResearch #012-000-003). Mice received two doses of 300 μg of anti-mouse NK1.1 (clone PK136 BioXCell #BE0036) or 300 μg of control mouse IgG (Jackson ImmunoResearch #015-000-003) intraperitoneally (IP) two days prior to ACT and on the day of ACT. Tumor volume was calculated as follows: $\text{Volume} = \frac{(\text{Length} \times \text{Width}^2)}{2}$.

**Flow cytometry**. Flow cytometry was performed on cultured cells or freshly explanted spleens, blood, tumors, livers, lungs, lymph nodes, or bone marrow. Tumors, livers, and lungs were weighed and treated with Liberase TL (Roche) and DNAse I (Sigma) for 30–45 min at 37 °C. Thirty milligrams of tumor or other tissue was stained and run on the flow cytometer. One hundred microliters of blood collected by submandibular vein bleed was subjected to red blood cell lysis and stained. With the exception of cells stained with Annexin V which were resuspended in Annexin V binding buffer (Biolegend), samples were fixed in 4% formaldehyde, and analyzed using the BD FACSCantoX SORP flow cytometer using FACSDiva v8.0 software or the ZE5 Cell Analyzer using Everest v2.0 softwawre in the Mayo Clinic Flow Cytometry Core. Mouse cells were stained with fluorochrome-conjugated antibodies against combinations of the following antigens: CD3 (Biolegend #100236 clone 145-2C11, dilution 1:500), CD8α (Biolegend #100738/100747, clone 53-6.7, dilution 1:1000), CD8β (Biolegend #140410, clone 53-5.8, dilution 1:500), CD4 (Biolegend #100451 clone GK1.5), Thy1.1 (Biolegend #202524, clone OX-7, dilution 1:1000; or eBioscience #11-0900-85, cloneH1S51, dilution 1:500), CD19 (biolegend #115537, clone 6D5, dilution 1:500), NK1.1 (Biolegend 108706/7, clone PK136, dilution 1:100), CXCR3 (Biolegend #126515, clone CXCR3-173, dilution 1:100), IFNAR1 (Biolegend #127314, clone MAR1-5A3, dilution 1:100), PD1 (Biolegend #109110, clone RMP1-30, dilution 1:200), LAG3 (Biolegend #125212, clone C9B7W, dilution 1:200), TIM3 (Biolegend #119704, clone RMT3-23, dilution 1:200), CD25 (Biolegend #102016, clone PC61 or BD #553866, clone PC61, dilutions both 1:200), CD69 (Biolegend #104513, clone H1.2F3, dilution 1:200), CD45 (Biolegend #103114, clone 30-F11, dilution 1:500), CD45.1 (Biolegend #110728/9, clone A20, dilution 1:250), CD45.2 (Biolegend #109828, clone 104, dilution 1:250), PD-L1 (Biolegend #124314, clone 10F.9G2, dilution 1:200), Fas (Biolegend #152604/152612, clone SA367H8, dilution 1:200), cleaved caspase (BD Pharmingen #51-68655X, clone C92-605). Transduction efficiency was determined using the Thy1.1 marker as well as biotinylated protein L (GenScript, dilution 1:100) and streptavidin–phycoerythrin, SA-PE (BD Biosciences #554061, dilution 1:100) or streptavidin– allophycocyanin, SA-APC (Invitrogen #1932748) as previously described[55]. Cell viability was determined using the Zombie fixable live dead viability dye (Biolegend #423106, dilution 1:1500) or Annexin V (Biolegend #640950, dilution 1:50). Cell proliferation in vivo was evaluated using CellTrace Violet (Thermo Fisher). Human T cells were stained with fluorochrome-conjugated antibodies against combinations of the following antigens: CD4 (Biolegend # 317433, clone OKT4, dilution 1:250), CD8 (Biolegend #300914, clone HIT8a, dilution 1:250), PD1 (Biolegend #329923, clone EH12.2H7, dilution 1:50), Tim3 (Biolegend #345015, clone F38-2E2, dilution

1:50), and Lag3 (Biolegend #369319, clone 11C3C65, dilution 1:50) and cleaved caspase 3 (R& D systems # IC835G, clone 269518, dilution 1:100). Data were analyzed using FlowJo version 10.5. A sample gating strategy for murine and human CAR T cells is shown in Supplementary Fig. 10.

**Cytokine/chemokine quantification**. The cytokine/chemokine array was performed on the supernatant derived from homogenized tumor lysate using the Legendplex Mouse Anti-Virus Response Panel (Biolegend). Data were analyzed using the Legendplex software 7.1.0.0. Interferon β in the tumor was quantified from homogenized tissue using the Verikine Mouse IFN Beta ELISA Kit (PBL Assay Science) and concentration in ng/mL was normalized to the average tumor weight.

**Immunofluorescence staining**. Seven micrometers cryosections were prepared from frozen tumors embedded in OCT, air dried at room temperature and fixed with ice-cold acetone for 10 min. Sections were blocked with 4% BSA in 0.1% Triton X-100 in PBS then with Vector M.O.M. Blocking Reagent (Vector Laboratories). Blocked sections were incubated overnight at 4 °C with anti-EGFRvIII (L8A4; Absolute Antibody) antibody diluted (1ug/mL) in 5% BSA. Sections were washed in PBS and incubated with secondary anti-mouse IgG1 AF488 (Biolegend # 406625, clone RMG1-1, dilution 1:100) antibody for 1 h at room temperature. Sections were incubated with 4′,6-diamidino-2-phenylindole (DAPI) diluted in PBS for 5 min at room temperature to counterstain nuclei washed in PBS. Sections were mounted with ProLong Gold Diamond Antifade (ThermoFisher). Images were captured using a Zeiss LSM780 microscope and analyzed using Zeiss's Zen software Black edition v8.0.

**Statistical analyses**. Data processing was performed in Microsoft Excel 2011. Graphing was performed with GraphPad Prism 6 software and statistical analysis was performed with GraphPad Prism 8 software (Graphpad). Single comparisons were made using unpaired two tailed t tests. Multiple comparisons were analyzed using one-way or two-way ANOVAs with a Tukey's or Sidak post-hoc multiple comparisons test. Survival data were assessed using the Log-Rank test, and P-values have not been adjusted for multiple testing. Data are expressed as group mean ± SD. Statistical significance set at $p < 0.05$ and ns > 0.05. Figures were prepared using Adobe Illustrator CC 2017.

**Reporting summary**. Further information on research design is available in the Nature Research Reporting Summary linked to this article.

## Data availability

All data generated during this study are available from the corresponding author upon reasonable request. The source data underlying all main and supplementary figs. are provided as a Source Data file. Animals were randomized to treatment groups using the GraphPad QuickCalcs online tool (https://www.graphpad.com/quickcalcs/randMenu/).

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

## Acknowledgements

The authors thank Steven A. Rosenberg, Richard A. Morgan, and Steven Feldman of the Surgery Branch at the National Cancer Institute for providing us with the EGFRvIII CAR retroviral construct. The authors thank Toni L. Woltman for expert secretarial assistance. This work was funded in part by Fraternal Order of Eagles Cancer Research Fund, Fellowship Program (to L. Evgin), The European Research Council (to R.G. Vile), The Richard M. Schulze Family Foundation (to R.G. Vile), the Mayo Foundation (to R.G. Vile), the NIH (R01CA175386, R01CA108961, P50 CA210964 to R.G. Vile), a grant from Terry and Judith Paul (to J.S. Pulido), and a research grant from Oncolytics Biotech Inc (to R.G. Vile). The salary of A. Huff and C. Driscoll was supported in part by grant T32 AI132165 from the National Institutes of Health.

## Author contributions

Conception and design: L.E., L.S.P., R.G.V. Development of methodology: L.E., L.S.P. Acquisition of data: L.E., A.L.H., P.W., T.K., K.G.S., J.T. Analysis and interpretation of data: L.E., A.L.H., T.K., C.B.D., M.S., K.A., P.R., D.M., A.J.J., J.P., L.S.P., R.G.V. Writing, review, and/or revision of the manuscript: L.E., L.S.P., R.G.V., S.Y. Administrative, technical, or material support: M.C., G.A., J.S. Study supervision: L.S.P., R.G.V.

## Competing interests

L.E., L.S.P., R.G.V. have intellectual property related to the use of IFNAR disruption in CAR T cells for combination with oncolytic viruses. M. Coffey is President and CEO at Oncolytics Biotech Inc and has ownership interest (including stock, patents, etc.) in Oncolytics Biotech Inc.
