## [Peer Review File · Nature Communications]

Reviewers' comments:

Reviewer #1 (Remarks to the Author):

Oncolytic viruses are being investigated in numerous laboratories as adjuvants to CAR T cell therapy. They are generally assumed to promote CAR T cell recruitment and function by generating inflammation through tumour-selective infection. Their potential combination is deemed very exciting and clinical trials are currently being discussed/planned.

In this manuscript, Evgin and colleagues present important and timely data that challenge the commonly-held and simplistic view of OV and CAR T cell combination therapy. Using the B16 syngeneic mouse model of melanoma, they show that oncolytic VSV and to a lesser extent oncolytic reovirus treatment do not benefit EGFR-CAR T cell therapy, as was predicted. Mechanistically, they provide evidence that type I IFN secreted in response to OV infection engenders CAR T cell dysfunction and promotes apoptosis *in vitro* and *in vivo*. Finally, they report that CRISPR-edited CAR T cells rendered deficient in IFNAR1 are resistant to IFN-mediated dysfunction and modestly more effective at clearing tumours *in vivo*. Collectively, this study is important because 1.) it suggests that combining OV with CAR T therapy needs to be done thoughtfully, taking into account the biology of the OV being used, the dosing regimen, and probably other factors (e.g. route of administration); 2.) it shows that IFNAR KO CAR T cells are partially resistant to the detrimental effects of OV-induced "heat".

Major Issues:

1. The authors must show that the mechanism holds true in human CAR T cells. At the very least, *in vitro* experiments testing IFN-mediated dysfunction and apoptosis in human CAR T cells must be performed. This is critically important given other work showing a strong positive role for type I IFN signaling in human CAR T cell biology. For example, Zhao et al 2015 Cancer Cell reported that IRF7-deficiency renders CAR T cells less able to clear NALM6 tumours *in vivo*. If IFNAR-deficiency in human CAR T cells has the opposite effect, as would be predicted from the author's mouse studies, some insights/thoughts as to why must be provided.
2. The modest effect of IFNAR-deficiency on CAR T cell function is concerning. Have the authors examined tumours for EGFR expression throughout treatment? I wonder if EGFR expression is partially (or completely) lost with treatment (i.e., antigen-loss relapse?). This would indicate a much stronger effect of IFNAR-deficiency, essentially showing that IFNAR-deficient CAR T cells are doing all they can.
3. It will be important to show that that the phenotype extends beyond a single tumour type. Key experiments should be performed in at least one other syngeneic model system.

Minor Issues:

1. It's not clear how well IT-injected VSV infects B16 tumours (which are IFN-competent and quite resistant to virus infection). Are the virus titers presented indicative of virus replication, or residual virus from the inoculation? Perhaps a couple earlier timepoints (e.g., 5 and 60 min) could be done to assess this.
2. Flow cytometry of lymphocytes should be done to demonstrate degree of lymphodepletion with IR. Without it, statements such as "VSVmIFNb mediated attrition could not be overcome by lymphodepletion either through depleting regulatory T cells or ..." should be revised.
3. Is it possible that VSV infects CAR T cells *in vivo* but not *in vitro*? Because of different activation state?

4. Is there any evidence that type I IFN (or other factors secreted by VSV infection) can alter the expression of T cell surface markers used for flow-based T-cell identification, which could give a false impression of T cell attrition?
5. Have the authors measure VSV infection in the spleen? Could it have leaked out of the tumour after injection and infected CD169 (or other) cells in the spleen? And if so, could that infection cause the CAR T cell attrition observed in the spleen?
6. Unless the authors show that Reovirus-mediated CAR T cell dysfunction is working through the same mechanism as VSV, the observational reovirus data should be moved to Supplemental Figures. And also, why were activation assays performed with Reovirus treatment (Fig. 3G) but not VSV?
7. Why were experiments in Fig. 4A-D performed 18 hours after CAR T cell transfer, while all previous experiments done 72 hours after? Please justify. As well as for the subsequent experiments in Fig. 4 that were performed 48 hours after transfer.
8. Why were experiments in Fig 5 (except those done at the bottom) done using a Jak/Stat inhibitor instead of IFNAR-deficient CAR T cells? The genetic approach is much cleaner.
9. Was there a positive selection for CRISPR-edited T cells? The efficiency is really high.
10. The Fig 1 label says "chemokine gradient", which was not measured. Intratumour chemokine levels is more accurate.
11. The flow plot in Fig 4H (VSVmIFN IT, no preconditioning, spleen) does not seem representative of the data presented in 4I.

Reviewer #2 could not submit a report.

Reviewer #3 (Remarks to the Author):

The manuscript: „Oncolytic virus derived type I interferon restricts CAR T cell therapy“ by Evgin et al. is a highly interesting study which deals with the important questions how different immunotherapies interfere with each other. Understanding interactions between different immune therapeutic approaches is one key aspect in improving immune-based tumor therapy. In this particular study the authors analyze the impact of the highly oncolytic VSVIFN β on CAR T cells. Analyzing such interaction is new and of high interest for the broad readership of Nature Communications. The authors found that VSVIFN β did not act synergistically to CAR T cell therapy, but rather led to reduction of the efficiency of CAR T cells. While VSVIFN β induced proliferation of CAR-T cells it induced apoptosis in T cells within the tumor. This apoptosis depended partially on Interferon type I (IFN-I), but as the authors claim it is very likely that additional factors, maybe synergistically to IFN-I, contribute to this T cell depletion. To prove the VSV-specificity in their model, the authors analyzed oncolytic Reovirus, and found that in fact also Reovirus infection limited CAR T cells survival in the tumor, although to a lesser degree.

In conclusion, the authors suggest that combination of oncolytic virus with CAR T cells might not lead to improvement. Their data argue against a postulated benefit of such a combination (PMID: 31383406). This is a very interesting piece of work. The major and minor points should be addressed before the manuscript can be considered for publication.

Major points:

IFN-I acts on several parameters which modulate T cell mediated cytotoxicity. On the one hand, IFN-I promotes T cell infiltration (15923172) and enhances antigen-presentation on target cells (15654326), which has a beneficial impact on T cell cytotoxicity. Further it protects T cells from NK cell mediated killing (24909889). On the other hand it accelerates expression of checkpoints and limits Proliferation of CD8+ T cells (23580529). The authors found accelerated death of CD8+ T cells in a IFN-I dependent manner. However even IFNAR-deficient T cells showed a remarkable reduction of CAR T cells in the tumor, suggesting that another mechanism might be more prominent in reducing CAR T cells in the tumor. Do other VSV-induced mechanisms act synergistically to CAR mediated cell death? Is PD-1L and IL-10 regulated by VSV within the tumor? The authors should analyze this.

The authors also found that VSV does not only limit the function of CAR T cells, but also PMEL CD8+ T cells. This is in contrast to other literature, which shows that combination of checkpoint blockade with Reovirus, VSV, T-vec or LCMV does have synergistic effects (PMID: 28886381, 26409567, 30972741, 27779616, 28248314), suggesting an improved CD8+ T cell function by these viruses. The authors should give some explanation how they would interpret this discrepancy. Does combination of VSV with CAR T cells and checkpoint blockade turn the unresponsiveness into synergistic effects? Or is the discrepancy explained by the state of differentiation of the T cells (naive, memory, effector,...)? Are there other possible explanations? The authors should clarify.

The authors found that Reovirus showed less negative effects on CAR T cells than VSV. VSV is highly cytotoxic while T-vec and also Reovirus might show a slower direct cytotoxic effect. Could the viral cytotoxicity of VSV in tumor cells or the viral cytotoxicity on CAR T cells modulate the survival of CAR T cells? Is there a tumor model where VSV and/or Reovirus shows much less or highly enhanced propagation? Would the influence on CAR T cells be similar in such a model?

Minor points:

How would bystander T cells or tumor-unspecific CAR T cells behave during VSV treatment?

How would virus specific T cells behave in the tumor, compared to spleen and lymph node?

NCOMMS-19-33777

Reviewer #1 (Remarks to the Author):

Oncolytic viruses are being investigated in numerous laboratories as adjuvants to CAR T cell therapy. They are generally assumed to promote CAR T cell recruitment and function by generating inflammation through tumour-selective infection. Their potential combination is deemed very exciting and clinical trials are currently being discussed/planned.

In this manuscript, Evgin and colleagues present important and timely data that challenge the commonly-held and simplistic view of OV and CAR T cell combination therapy. Using the B16 syngeneic mouse model of melanoma, they show that oncolytic VSV and to a lesser extent oncolytic reovirus treatment do not benefit EGFR-CAR T cell therapy, as was predicted. Mechanistically, they provide evidence that type I IFN secreted in response to OV infection engenders CAR T cell dysfunction and promotes apoptosis *in vitro* and *in vivo*. Finally, they report that CRISPR-edited CAR T cells rendered deficient in IFNAR1 are resistant to IFN-mediated dysfunction and modestly more effective at clearing tumours *in vivo*. Collectively, this study is important because 1.) it suggests that combining OV with CAR T therapy needs to be done thoughtfully, taking into account the biology of the OV being used, the dosing regimen, and probably other factors (e.g. route of administration); 2.) it shows that IFNAR KO CAR T cells are partially resistant to the detrimental effects of OV-induced “heat”.

Major Issues:

1. The authors must show that the mechanism holds true in human CAR T cells. At the very least, *in vitro* experiments testing IFN-mediated dysfunction and apoptosis in human CAR T cells must be performed. This is critically important given other work showing a strong positive role for type I IFN signaling in human CAR T cell biology. For example, Zhao et al 2015 Cancer Cell reported that IRF7-deficiency renders CAR T cells less able to clear NALM6 tumours *in vivo*. If IFNAR-deficiency in human CAR T cells has the opposite effect, as would be predicted from the author’s mouse studies, some insights/thoughts as to why must be provided.

We thank the reviewer for their critical analysis of our work and their insightful questions. We have added new **Figure 5** in which we have generated human EGFRvIII CAR T cells and cultured them *in vitro* for 48 hours with various doses of recombinant IFN β , analogously to our mouse experiments. Both CD4 and CD8 untransduced and CAR T cells exhibited a dose dependent increase in cleaved caspase 3 staining, demonstrating that the virus-associated apoptosis observed in mice is consistent with human T cells. We have also examined inhibitory receptor expression and seen an increase in PD1, Tim3 and Lag3, particularly at high doses of IFN β . We have thus added the following text to the results and discussion sections, on pages 9-10 and 11, respectively:

“In order to validate the murine findings in human cells, EGFRvIII CAR T cells were generated from healthy donor PBMCs using a retroviral construct encoding the same targeting scFv and the corresponding human signaling domains (30). Analogous to **Figure 4**, untransduced or CAR T cells from three independent donors were cultured *in vitro* in the presence of recombinant human IFN β for 48 hours. We observed a dose dependent increase in apoptosis as measured by cleaved caspase 3 staining in both CD4 and CD8 CAR transduced and untransduced T cells (**Fig 5A**). Consistent with the mouse studies as well, we also observed elevated expression of the inhibitory receptors PD1, TIM3, and in particular, LAG3, in the presence of increasing concentrations of IFN β (**Fig 5B-E**).”

“Both *in vivo* and *in vitro*, CAR T cells which were exposed to type I IFN were sensitized to apoptosis and displayed greater activation marker and inhibitory receptor expression. Furthermore, IFN β -induced T cell apoptosis and inhibitory receptor expression were confirmed to also occur in human CAR and untransduced T cells.”

There are two important differences between our work and that described by Zhao et al in their Cancer Cell paper. The authors explore different configurations of costimulation where the receptor is either incorporated in the CAR molecule (1928BBz) or where the ligand is co-expressed (1928z-41BBL). T cells transduced with these different configurations behaved very differently in terms of their ability to persist and eradicate tumor *in vivo*. Our CAR configuration closely resembles the 1928BBz and not the 1928z-41BBL, the latter of which is specifically associated with the beneficial IFN β signaling profile. Secondly, the concentration of IFN β reported to be produced following antigen stimulation of the 1928z-41BBL CAR in Fig 7B is approximately 10 pg/mL. This is at least 100-1000 fold lower than the concentration that we observe to be induced by virus infection in the B16EGFRvIII VSVmIFN β infected tumors (1-50 ng/mL), and which we modeled *in vitro*. Thus, we reason that the consequence of IFN β exposure is not only dependent on antigen stimulation and is potentiated by cis/ trans-costimulation, but that it is context and concentration dependent, as is well known for type I IFNs.

We have added the following text to the discussion on page 13 to address this point:

“It has been previously reported that a type I IFN gene signature has been associated with enhanced functionality of a CAR configuration where 41BB-L co-stimulation is co-expressed rather than incorporated into the CAR molecule (41). Not only has this particular signature only been reported for the 1928z-41BBL configuration and not the 1928BBz which more closely resembles our construct, the reported concentration of IFN β was 2-3 orders of magnitude lower than that induced by VSV. Thus, we reason that the consequence of IFN β exposure is not only dependent on antigen stimulation and is potentiated by cis/ trans-co-stimulation, but that it is context and concentration dependent, as is well known for type I IFNs.”

2. The modest effect of IFNAR-deficiency on CAR T cell function is concerning. Have the authors examined tumours for EGFR expression throughout treatment? I wonder if EGFR expression is partially (or completely) lost with treatment (i.e., antigen-loss relapse?). This would indicate a much stronger effect of IFNAR-deficiency, essentially showing that IFNAR-deficient CAR T cells are doing all they can.

We have stained endpoint tumors which have escaped therapy with CAR T cells and observed levels of EGFRvIII which were similar to tumors which received mock treatment. Some regions of CAR T cell treated tumors exhibit reduced levels of staining, while others do not. Although we have not identified the factors which lead to escape from CAR T therapy in this model, we do not hypothesize that complete antigen escape is the primary mechanism. We have provided the images of the staining in **Supplementary Fig 2** and added the following the text to the results section on page 5.

“EGFRvIII expression was largely maintained in B16EGFRvIII tumors which escaped CAR T therapy, suggesting that antigen loss is not a primary mechanism of therapeutic failure in this model.”

3. It will be important to show that that the phenotype extends beyond a single tumour type. Key experiments should be performed in at least one other syngeneic model system.

We have evaluated VSV-associated attrition in a second syngeneic tumor model, CT2AEGFRvIII, and have added this data to **Supplementary Fig 3**. When injected with VSVmIFN β , infectious virus persists for at least 48h and IFN β is detected in the tumor at a concentration of approximately 1-10 ng/mL. The level of transgene is not as high as the B16EGFRvIII tumors, although it is in the same range. Importantly, when we injected VSVmIFN β either 6 or 24 hours prior to adoptive cell transfer (ACT), we observed a significant reduction in the number of viable CD8 CAR T cells recovered from the tumors relative to a PBS injection. Furthermore, a trend toward CD4 CAR T cell attrition was also observed in the tumor, although it did not reach statistical significance. There was no reduction in the number of CAR T cells recovered from the spleen. Similar to the B16EGFRvIII model, there was no impairment in CAR T proliferation in the tumor when VSVmIFN β was injected 6 or 24h prior to ACT which may account for the reduced number of recovered cells. We have added the following text to the results section on page 6.

“VSVmIFN β - associated attrition was also evaluated in a subcutaneous CT2AEGFRvIII tumor model which supports VSV infection and transgene expression (**Supplemental Fig 3 A-I**). As observed in the B16EGFRvIII tumors, when CT2AEGFRvIII tumors were intratumorally treated 6 or 24 hours prior to adoptive transfer, significantly fewer CD8 CAR T cells were recovered compared to a PBS injection. A trend toward CD4 CAR T cell attrition was also observed in the tumor, although it did not reach statistical significance. Cell trace violet dilution also did not reveal impaired proliferation as a mechanism which would account for the reduction in CAR T cell number (**Supplemental Fig 3 J,K**).”

Minor Issues:

1. It's not clear how well IT-injected VSV infects B16 tumours (which are IFN-competent and quite resistant to virus infection). Are the virus titers presented indicative of virus replication, or residual virus from the inoculation? Perhaps a couple earlier timepoints (e.g., 5 and 60 min) could be done to assess this.

The VSVmIFN β titer in the B16EGFRvIII tumors is shown in **Figure 1C**. As the reviewer suggests, the virus detected 6 hours post-injection could be either indicative of virus replication, or residual from the inoculation. However, since the titer increases between 6 hours and 24 hours post injection, we would argue that there is a modest level of virus replication. Although *in vitro* and *in vivo* infectivity can be different, we have additionally added single and multi-step *in vitro* growth curves into **Supplemental Fig 1B** which is consistent with the B16EGFRvIII tumors supporting a modest level of replication *in vivo*.

The text on page 5 was modified as follows:

“IFN α was also induced along with a variety of inflammatory chemokines including CXCL10, and CCL5, and their expression mirrored the kinetics of viral replication in the tumor *in vivo* and *in vitro* (**Fig 1B,C, Supplementary Fig 1B**).”

2. Flow cytometry of lymphocytes should be done to demonstrate degree of lymphodepletion with IR. Without it, statements such as “VSVmIFN β mediated attrition could not be overcome by lymphodepletion either through depleting regulatory T cells or ...” should be revised.

We have added new data to **Supplementary Figure 1D** quantifying the T cell, B cell and NK cell compartments in the blood 2 days post irradiation to validate the lymphodepletion. We have added the following text to page 5 in the Results:

“We repeated the same schedule of VSVmIFN β and EGFRvIII CAR T cell mono- or combination therapy in mice that received a lymphodepleting dose of 5 Gy whole body irradiation to fully capture the therapeutic potential of the CAR T cells (**Fig 1G**). Lymphodepletion was confirmed in submandibular bleeds 2 days post-irradiation (Supplemental Fig 1D).”

We have additionally modified the sentence in question on page 5 to remove the inference of specifically removing T reg populations and inducing homeostatic cytokines.

“Similarly, fewer CD8 CAR T cells were detected in the blood of mice which received a single dose or three doses of VSVmIFN β , suggesting that the VSVmIFN β mediated attrition could not be overcome by lymphodepletion ~~either through depleting regulatory T cells or the induction of homeostatic pro-survival cytokines (Fig 1I).~~”

3. Is it possible that VSV infects CAR T cells *in vivo* but not *in vitro*? Because of different activation state?

As the VSV-associated attrition occurs very rapidly following adoptive transfer in VSV infected animals, we do not hypothesize that *in vivo* conditioning and differentiation is required to sensitize the T cells to apoptosis. We therefore reason that our *in vitro* assay is a reasonable approximation of the sensitivity of the T cells to infection. However, given our dataset, it is not possible to rule out that the level of infection of CAR T cells may be different *in vitro* and *in vivo*.

We have modified the text on page 5 as follows:

We confirmed that VSVmIFN β does not directly promote apoptosis of CAR T cells *in vitro* (Supplemental Fig 1E), however, it is not possible to rule out that the level of infection of CAR T cells may be different *in vitro* and *in vivo*.

4. Is there any evidence that type I IFN (or other factors secreted by VSV infection) can alter the expression of T cell surface markers used for flow-based T-cell identification, which could give a false impression of T cell attrition?

We have retroviral constructs which express both Thy1.1 and GFP from an IRES and we have observed CAR T cell attrition in mice following VSVmIFN β injection using both these systems. *In vitro*, when CAR T cells or untransduced T cells are treated with IFN β , we do not observe a reduction in the expression of CD8 or CD4 that would render the cells incapable of being detected.

5. Have the authors measure VSV infection in the spleen? Could it have leaked out of the tumour after injection and infected CD169 (or other) cells in the spleen? And if so, could that infection cause the CAR T cell attrition observed in the spleen?

It is likely that some virus leaks out of the tumor following an intratumoral injection, however we have not directly measured VSVmIFN β titers or transgene expression or type I IFN induction in the spleen. It has been reported that WT and oncolytic attenuated VSV can replicate to some degree in the spleen following intravenous administration (PMID: 22101728 and 30073187 respectively) and in particular CD169+ subcapsular sinus macrophages are amongst the most permissive cells. Based on our data, we cannot distinguish whether there is CAR T cell apoptosis occurring directly in the spleen due to low level infection, or whether the more modest level of attrition simply reflects apoptosis which occurred in the tumor and resulted in reduced levels of recirculating CAR T cells.

The following text was added to page 6:

“The loss of CAR T cells was more profound in the tumor than the spleen. Based on these data, it is not possible to distinguish whether the levels of CAR T cells in the spleen reflected recirculated cells from the tumor, or whether some virus infection associated apoptosis directly occurred in the spleen.”

6. Unless the authors show that Reovirus-mediated CAR T cell dysfunction is working through the same mechanism as VSV, the observational reovirus data should be moved to Supplemental Figures. And also, why were activation assays performed with Reovirus treatment (Fig. 3G) but not VSV?

We have moved the reovirus data previously in **Figure 3** to **Supplemental Fig 5**.

The activation status of the CAR T cells following VSVmIFN β has been added to **Supplemental Fig 4** and the following text has been added to the results section on page 7.

“Increased activation of CD8 and CD4 CAR T cells was also observed in the spleen 24 hours after adoptive transfer into VSVmIFN β infected mice (**Supplemental Fig 4**).”

7. Why were experiments in Fig. 4A-D performed 18 hours after CAR T cell transfer, while all previous experiments done 72 hours after? Please justify. As well as for the subsequent experiments in Fig. 4 that were performed 48 hours after transfer.

The time point in **Fig 4A-D** was chosen to quantify the maximum level of IFN β produced in the tumor following VSVmIFN β or VSVGFP injection (see kinetics in **Fig 1C**) and to correlate this with the number of recovered CAR T cells. The 18 hour post adoptive transfer time point was an additional 6 hours after VSV injection, totaling 24 hours post virus injection. These data are also consistent with **Figure 2K** and **Supplemental Fig 4** which shows high level attrition 24 hours post injection of CAR T cells.

8. Why were experiments in Fig 5 (except those done at the bottom) done using a Jak/Stat inhibitor instead of IFNAR-deficient CAR T cells? The genetic approach is much cleaner.

We began the *in vitro* IFN β culture experiments prior to using the CRISPR approach. As we do have the genetic KO (both transgenic in **Fig 5M/N** and CRISPR modified cells in **Fig 6B-D**) and the pharmacological blockade approaches, we feel that that these two datasets provide complementary information.

9. Was there a positive selection for CRISPR-edited T cells? The efficiency is really high.

There was no positive selection for the CRISPR modification. We have reproducibly generated high level genetic KOs using the cited protocol (PMID: 29436394).

10. The Fig 1 label says “chemokine gradient”, which was not measured. Intratumour chemokine levels is more accurate.

We have modified the text to replace all reference of “chemokine gradient” to “chemokine profile.” This was changed in the introduction on page 4, the results on page 5, the discussion on page 11, and in the Figure 1 legend on page 20.

11. The flow plot in Fig 4H (VSVmIFN IT, no preconditioning, spleen) does not seem representative of the data presented in 4I.

The particular data point referenced is the middle of the three values. We have provided the raw data below and indicated in blue which of the mice were used for the representative **Fig 4H**.

Percent of CD8 CAR:		WT	IFNAR KO	Ratio	Normalized ratio
	Input CD8	52.4	47.5	0.91	
SPLEEN	PBS IT 1	56	43.7	0.78	0.86
	PBS IT 2	56.1	43.7	0.78	0.86
	PBS IT 3	54.8	45	0.82	0.91
	PBS IT 4	54.3	45.5	0.84	0.92
	VSV IT 1	53.7	45.9	0.85	0.94
	VSV IT 2	50.9	48.9	0.96	1.06
	VSV IT 3	50.6	49.2	0.97	1.07
	TBI PBS IT 1	52.1	47.6	0.91	1.01
	TBI PBS IT 2	52.3	47.5	0.91	1.00
	TBI PBS IT 3	54.3	45.5	0.84	0.92
	TBI VSV IT 1	45.8	54	1.18	1.30
	TBI VSV IT 2	41.7	58.1	1.39	1.54
	TBI VSV IT 3	41.2	58.7	1.42	1.57
TBI VSV IT 4	39.8	60	1.51	1.66	
TUMOR	PBS IT 1	46.8	52.9	1.13	1.25
	PBS IT 2	47.4	52	1.10	1.21
	PBS IT 3	44.6	55.1	1.24	1.36
	PBS IT 4	43.7	55.9	1.28	1.41
	VSV IT 1	37.6	62.1	1.65	1.82
	VSV IT 2	35.3	64.4	1.82	2.01
	VSV IT 3	30	69.7	2.32	2.56
	TBI PBS IT 1	39.9	59.8	1.50	1.65
	TBI PBS IT 2	40	59.4	1.49	1.64
	TBI PBS IT 3	40.2	59.4	1.48	1.63
	TBI VSV IT 1	31.2	68.5	2.20	2.42
	TBI VSV IT 2	29.7	69.7	2.35	2.59
	TBI VSV IT 3	27.5	72.2	2.63	2.90
TBI VSV IT 4	29.9	69.6	2.33	2.57	

Reviewer #3 (Remarks to the Author):

The manuscript: "Oncolytic virus derived type I interferon restricts CAR T cell therapy" by Evgin et al. is a highly interesting study which deals with the important questions how different immunotherapies interfere with each other. Understanding interactions between different immune therapeutic approaches is one key aspect in improving immune-based tumor therapy. In this particular study the authors analyze the impact of the highly oncolytic VSVIFN β on CAR T cells. Analyzing such interaction is new and of high interest for the broad readership of Nature Communications.

The authors found that VSVIFN β did not act synergistically to CAR T cell therapy, but rather led to reduction of the efficiency of CAR T cells. While VSVIFN β induced proliferation of CAR-T cells it induced apoptosis in T cells within the tumor. This apoptosis depended partially on Interferon type I (IFN-I), but as the authors claim it is very likely that additional factors, maybe synergistically to IFN-I, contribute to this T cell depletion.

To prove the VSV-specificity in their model, the authors analyzed oncolytic Reovirus, and found that in fact also Reovirus infection limited CAR T cells survival in the tumor, although to a lesser degree.

In conclusion, the authors suggest that combination of oncolytic virus with CAR T cells might not lead to improvement. Their data argue against a postulated benefit of such a combination (PMID: 31383406). This is a very interesting piece of work. The major and minor points should be addressed before the manuscript can be considered for publication.

Major points:

IFN-I acts on several parameters which modulate T cell mediated cytotoxicity. On the one hand, IFN-I promotes T cell infiltration (15923172) and enhances antigen-presentation on target cells (15654326), which has a beneficial impact on T cell cytotoxicity. Further it protects T cells from NK cell mediated killing (24909889). On the other hand it accelerates expression of checkpoints and limits proliferation of CD8+ T cells (23580529). The authors found accelerated death of CD8+ T cells in a IFN-I dependent manner. However even IFNAR-deficient T cells showed a remarkable reduction of CAR T cells in the tumor, suggesting that another mechanism might be more prominent in reducing CAR T cells in the tumor. Do other VSV-induced mechanisms act synergistically to CAR mediated cell death? Is PD-1L and IL-10 regulated by VSV within the tumor? The authors should analyze this.

We thank the reviewer for their critical analysis of our work and their insightful questions. As has been pointed out, type I IFN induces a myriad of changes in the tumor microenvironment on a variety of different cell types. Even though IFNAR KO CAR T cells have a selective advantage over WT CAR T cells in non-preconditioned mice, there is still a significant level of attrition observed (**Fig 4J** left panel). However, the number of IFNAR KO CAR T cells is not significantly different between PBS and VSV injected tumors in irradiated mice (**Fig 4J** right panel). This suggested to us that IFN β -associated apoptosis is the primary mediator of VSV-associated attrition in the absence of a particular cell type that is depleted by radiation. It has been previously shown that type I IFN signaling is necessary to protect CD8 T cells against NK attack in the context of virus infection (PMID 24909889, 24909887). We therefore repeated the experiment in **Fig 3G-I** using the anti-NK1.1 antibody to deplete NK cells and observed that NK depletion functionally replicated the effect of TBI. Therefore, IFNAR KO CAR T cells are protected against the direct apoptosis induced by VSV derived type I IFN, however subsequently rendered sensitive to NK attack in non-lymphodepleted animals. This data has been added to the new **Supplemental Figure 6**.

The following text has been added to the results section on page 8:

“As it has been previously shown that type I IFN signaling is necessary to protect CD8 T cells against NK attack in the context of virus infection (26, 27), we hypothesized that the enhanced selection and recovery of IFNAR KO CD8 CAR T cells in the VSV-pretreated lymphodepleted animals compared to VSV-pretreated non-preconditioned animals may reflect the depletion of NK cells. Anti-NK1.1 antibody administration functionally reproduced the effect of TBI where the selective recovery of IFNAR KO CAR T cells was enhanced in the absence of NK cells (**Supplemental Fig 6**). Therefore, IFNAR KO CAR T cells are protected against the direct apoptosis induced by VSV derived type I IFN, however subsequently rendered sensitive to NK attack in non-lymphodepleted animals.”

These data point to a key role of direct type I IFN signaling on T cells in the tumor. However, many other molecules, such as PDL1, are upregulated by type I IFN on tumor cells

(Supplemental Figure 9B) or other immune cells and thus may indirectly adversely affect T cell survival or function. These types of changes are best characterized by a highly comprehensive approach such as RNA-seq or CYTOF and will form the basis of future studies.

The authors also found that VSV does not only limit the function of CAR T cells, but also PMEL CD8+ T cells. This is in contrast to other literature, which shows that combination of checkpoint blockade with Reovirus, VSV, T-vec or LCMV does have synergistic effects (PMID: 28886381, 26409567, 30972741, 27779616, 28248314), suggesting an improved CD8+ T cell function by these viruses. The authors should give some explanation how they would interpret this discrepancy. Does combination of VSV with CAR T cells and checkpoint blockade turn the unresponsiveness into synergistic effects? Or is the discrepancy explained by the state of differentiation of the T cells (naive, memory, effector,...)? Are there other possible explanations? The authors should clarify.

Indeed, various oncolytic viruses and vaccines have been shown to increase T cell infiltration into tumors, however this broad statement does not consider the (1) the time point of measurement (2) the dose and route of virus administration (3) the differentiation state of the T cells at the time of virus exposure and (4) the specificity of the T cells. For example, Ribas et al (PMID 28886381) compare T cell density between baseline lesions and a 6-week post injection time point. During this 6-week period, tumor resident T cells may have undergone apoptosis via the mechanism that we have described, and newly primed anti-viral or anti-tumor T cells may have replaced these cells. Similarly, Koske et al (PMID 30972741) and Kalkavan et al (PMID: 28248314) report increased T cell infiltration 7- and 10-days post virus administration, respectively. This is a highly dynamic process where the acute virus associated effects are not captured at a single late time point. Furthermore CD3/CD8/CD4 T cell staining by flow cytometry or IHC provides information about the density of the TIL population, but not about the specificity of the T cells, and critically, whether they are tumor-specific, virus-specific, or ignorant to both.

Prior studies from our lab (Cockle et al PMID: 26409567 and others) have used low-dose intravenously administered virus to prime a *de novo* T cell responses against virally encoded antigens. This type of systemic vaccination minimizes the local concentration of type I IFN in the tumor while still providing some low dose IFN in secondary lymphoid organs to potentiate the priming of naïve T cells against either viral or tumor associated antigens. Similarly, Ilett et al (PMID 27779616) employs a low systemic dose of VSV encoding tumor antigens to boost tumor reactive T cells. In this case, we would hypothesize that minimal virus replication occurs in the tumor, thereby limiting the local concentration of IFN β .

We have added the following text to the discussion on pages 12 to address this point:

“While high level IFN may facilitate the priming of tumor reactive T cells during oncolysis, it may also promote memory T cell attrition to make room for more diverse and effective T cell responses to novel pathogens (17). Although multiple preclinical and clinical reports have documented increased TIL density following virus treatment, we reason that several factors are necessary to interpret how an oncolytic virus and its associated inflammation may dynamically potentiate the number of T cells in a tumor (1) the time point of T cell enumeration relative to virus injection (2) the dose and route of administration (3) the differentiation state of the T cells at the time of virus exposure and (4) the specificity of the T cells.”

With respect to the second question as to whether the combination of VSV with CAR T cells and checkpoint blockade would turn the unresponsiveness into synergistic effects, we do not have evidence to support this suggestion. Indeed, we have shown that IFN β upregulates PDL1 and Galectin 9 in tumor cells (**Supplemental Figure 9B**). This finding led us to hypothesize that

concurrent checkpoint blockade of PD1, TIM3 and LAG3 would improve combination therapy using VSV and CAR T cells (**Supplemental Figure 9C**). Although we observed that checkpoint blockade modestly improved the efficacy of CAR T therapy, triple combination with VSVGFP led to further very modest increase in tumor control that was not synergistic.

The authors found that Reovirus showed less negative effects on CAR T cells than VSV. VSV is highly cytotoxic while T-vec and also Reovirus might show a slower direct cytotoxic effect. Could the viral cytotoxicity of VSV in tumor cells or the viral cytotoxicity on CAR T cells modulate the survival of CAR T cells? Is there a tumor model where VSV and/or Reovirus shows much less or highly enhanced propagation? Would the influence on CAR T cells be similar in such a model?

Whether secondary mediators induced by or in response to tumor cell oncolysis have deleterious effects on CAR T cell health is an interesting question. Both the magnitude and type of factors released would differ for VSV and Reovirus according to their lytic ability, replication kinetics etc. For example, it is certainly possible that there are pathogen- or danger- associated molecular pattern molecules which are released upon oncolysis which could adversely affect CAR T cell viability. As described above, a more comprehensive analysis of the ways in which viral oncolysis remodels the tumor microenvironment will form the basis of future work. To address this point, we have added the following text to the discussion on page 12.

“Although we have described a mechanism in which type IFN plays a major role in the therapeutic interference that we observed with VSV, it is certainly possible that there are other pathogen- or danger- associated molecular pattern molecules which are released upon oncolysis which could adversely affect CAR T cell health.”

Minor points:

How would bystander T cells or tumor-unspecific CAR T cells behave during VSV treatment?

In **Figure 2E**, we do observe a modest reduction in the endogenous CD8 T cell compartment when VSVmIFN β is injected 24 hours prior to ACT compared to PBS treatment. We have also observed increased activation marker expression (CD25 and CD69) on endogenous CD4 and CD8 in the tumor and spleen in VSVmIFN β infected mice (**Supplemental Fig 4**). This data, however, does not distinguish between endogenous T cells bearing a tumor specific or a tumor-ignorant TCR.

How would virus specific T cells behave in the tumor, compared to spleen and lymph node?

At the time of virus infection, all mice are naïve to VSV. Therefore, naïve T cells which have a TCR recognizing VSV epitopes would likely be located either in the spleen or lymph node, but not bear the necessary homing molecules to reach the tumor. T cell priming would then occur as antigen presenting cells take up viral material, traffic to lymphoid organs and present it to cognate T cells in the presence of low levels of type I IFN.

REVIEWERS' COMMENTS:

Reviewer #1 (Remarks to the Author):

The authors have sufficiently addressed my concerns. The manuscript is considerably stronger and suitable for publication. I have no further questions/comments.

Reviewer #3 (Remarks to the Author):

The authors addressed all my concerns with additional experiments or clear statements in the results or discussion section. It is a nice piece of work with a huge amount of cutting edge experiments. It will bring new important insights into the field of immune oncology and will certainly lead to fruitful to controversial discussions. I highly recommend the publication without further delay.